# Mapping global kimberlite potential from reconstructions of mantle flow over the past billion years

**Anton Grabreck, Nicolas Flament**[ID]**\*, Ömer F. Bodur**

GeoQuEST Research Centre, School of Earth, Atmospheric and Life Sciences, University of Wollongong, Wollongong, NSW, Australia

* nflament@uow.edu.au

## Abstract

Kimberlites are the primary source of economic grade diamonds. Their geologically rapid eruptions preferentially occur near or through thick and ancient continental lithosphere. Studies combining tomographic models with tectonic reconstructions and kimberlite emplacement ages and locations have revealed spatial correlations between large low shear velocity provinces in the lowermost mantle and reconstructed global kimberlite eruption locations over the last 320 Myr. These spatial correlations assume that the lowermost mantle structure has not changed over time, which is at odds with mantle flow models that show basal thermochemical structures to be mobile features shaped by cold sinking oceanic lithosphere. Here we investigate the match to the global kimberlite record of stationary seismically slow basal mantle structures (as imaged through tomographic modelling) and mobile hot basal structures (as predicted by reconstructions of mantle flow over the past billion years). We refer to these structures as "basal mantle structures" and consider their intersection with reconstructed thick or ancient lithosphere to represent areas with a high potential for past eruptions of kimberlites, and therefore areas of potential interest for diamond exploration. We use the distance between reconstructed kimberlite eruption locations and kimberlite potential maps as an indicator of model success, and we find that mobile lowermost mantle structures are as close to reconstructed kimberlites as stationary ones. Additionally, we find that mobile lowermost mantle structures better fit major kimberlitic events, such as the South African kimberlite bloom around 100 Ma. Mobile basal structures are therefore consistent with both solid Earth dynamics and with the kimberlite record.

## 1. Introduction

The majority of diamonds, including economic grade diamonds, were formed at the base of thick continental lithosphere [1]. Using xenolith data from kimberlitic eruptions, ref. [2] determined the temperature-pressure window that allows for diamond formation and stability as between 1200°C and 1570°C and above 5 GPa. "Clifford's rule" postulates that cratonic diamonds are formed in cratons with thick lithospheric keels, providing environments with sufficiently high pressures and low temperatures for diamond stability [3, 4].

LP170100863 (Industry partner: De Beers). This research was supported by the Australian Government's National Collaborative Research Infrastructure Strategy (NCRIS), with access to computational resources provided by the National Computational Infrastructure (NCI) through the National Computational Merit Allocation Scheme. Access to NCI was partly supported by resources and services from the University of Wollongong (UOW).

**Competing interests:** The authors have declared that no competing interests exist.

Cratons of Archean age are particularly thick [possibly up to ~300 km; e.g., 5] and are an ideal environment for diamond formation [6–8]. Kimberlite eruptions are rapid magmatic events with magma ascending at speeds up to 20 m s$^{-1}$ [9, 10], which is geologically instantaneous compared to average plate motion at ~$1.26 \times 10^{-9}$ m s$^{-1}$ (4 cm yr$^{-1}$) over the last 200 Myr [11]. Kimberlites originate from depths in excess of 120–150 km [12], forming from melts that may pool from up to ~300 km depth [13]. Some rare superdeep (sub-lithospheric) kimberlites may have ascended from as deep as 800 km [14–17], although these diamonds could have been transported to the base of the lithosphere before being entrained by kimberlite magmas [18]. The kimberlite record extends deep in geological times [the oldest known kimberlite pipe is 2.85 Gyr old; 19] and is considered to be episodic from 2 Ga [20–22]. Over 60% of known kimberlite pipes erupted during Mesozoic and Cenozoic times [22], which could be either the result of the preferential preservation of younger kimberlites or of increased eruption rate during the last 200 Myr [23]. Southern Africa underwent intensive kimberlite activity during this period, and this area preserves the most known kimberlite occurrences [12, 21, 22].

Combining databases of volcanic products with tectonic reconstructions and SMEAN [an average of three S-wave tomographic models; 24], ref. [25] showed that the reconstructed locations of the majority of large igneous provinces (LIPs) and kimberlites from 320 Ma (to the exception of Canadian kimberlites) are within ~1,000 km of the edges of two large seismically slow provinces above the core-mantle boundary (CMB) commonly known as large low-shear velocity provinces [LLSVPs; 26]. The LLSVPs, presently positioned under the Pacific Ocean (the Pacific LLSVP) and under the African continent (the African LLSVP), are ~7,500 km in radius and 500–1,000 km in height test–together they cover up to 50% of the CMB [26, 27]. The low shear-wave velocity of LLSVPs indicates that these structures are hotter than surrounding ambient mantle, and gradients in seismic waves at the edges of LLSVPs [28] suggest that the structures are intrinsically denser than surrounding mantle, although the density contrast remains debated [e.g., 29].

Due to their spatial relationship with volcanic surface features, the regions along the edge of LLSVPs are sometimes called plume generation zones (PGZs) [30]. The specific relationship between the edges of LLSVPs and reconstructed locations of volcanic products is debated, with studies showing that the relationship is instead between the interior of LLSVPs and reconstructed locations of volcanic products [31, 32]. It is generally accepted that major LIPs are linked to deep mantle plumes [33, 34], and the synchronicity of Mesozoic-Cenozoic kimberlite blooms with periods of LIP eruption suggests a potentially common heat source for kimberlite and LIP eruptions [25, 35]. However, the physical process connecting kimberlite magmatism [from 120–300 km depth; 12, 13] to LLSVPs [between 2000 km and 2500 km depth; 26] remains to be established. The spatial link between LLSVPs and reconstructed kimberlite eruption locations implies that hot basal mantle structures could have been stationary and rigid over time [25]. In contrast, global geodynamic models have revealed that similar hot basal mantle structures can form in response to the history of surface plate motions and plate subduction [36, 37].

Here we assume that kimberlite eruptions are linked to mantle upwelling occurring above basal mantle structures, following ref. [25]. We use the global tectonic reconstruction of ref. [38] to derive boundary conditions for forward mantle convection models that predict the evolution of the structure of the mantle over the past billion years. We map the intersection of basal mantle structures (either stationary from tomography or mobile from flow models) and reconstructed lithosphere, which represent areas with a high potential for past eruptions of kimberlites and are therefore of potential interest to diamond exploration. We use the distance between the reconstructed locations of known kimberlite eruptions and model high-potential

areas as an indicator of model success and evaluate the fit of different models to the kimberlite record for selected periods and regions of high kimberlite activity.

## 2. Datasets and methods

Our approach uses tectonic plate reconstructions and kimberlite emplacement ages to link past kimberlite eruption locations, thick or ancient lithosphere, and basal mantle structures either imaged by tomographic models or predicted by mantle flow models. Key model results and scripts are available at https://doi.org/10.5281/zenodo.5760115.

### 2.1. Tectonic reconstructions

We use the global tectonic reconstruction by ref. [38], which is the first continuous full-plate reconstruction for the last billion years. This tectonic reconstruction links full-plate reconstructions by ref. [39] from 1,000–520 Ma, by ref. [40, 41] from 500 Ma to 410 Ma and by ref. [42] from 410 to 0 Ma. We consider the tectonic reconstruction in its original paleomagnetic reference frame (reconstruction hereafter referred to as M21), as well as in a reference frame in which the net rotation of the lithosphere with respect to the mantle was removed, which is more appropriate for mantle flow models [e.g., 43, 44]. This latter reconstruction is hereafter referred to as M21NNR ("no-net-rotation").

### 2.2. Kimberlite locations and emplacement ages, and thick or ancient lithosphere

We use the kimberlite database of ref. [22] that contains 1,133 kimberlites with a peak in occurrences at around 100 Ma (Fig 1B)–notably in southern Africa (Fig 1A). Kimberlite occurrences were sampled at 20 Myr intervals to match the temporal resolution used for the output of global mantle flow models (see below). For each age of interest $a_n$, kimberlite occurrences were selected if they erupted within the period $a_n + 10$ Myr $< a_n \leq a_n - 10$ Myr.

The age of the lithosphere can be established locally based on xenoliths carried to the surface by kimberlites [e.g., 45], and continental scale maps require interpolation. We therefore consider three end-member sets of thick or ancient lithosphere: 1/ lithosphere thicker than 150 km (a proposed proxy for cratons, ref. [4]) in all four lithospheric thickness models from ref. [5]; 2/ regions with tectonothermal ages greater than 2.5 Ga in the TC1 model of ref. [46] and 3/ tectonic blocks inferred to have existed for at least one billion years from ref. [38]. Almost all present-day locations of kimberlites fall within the tectonic blocks from ref. [38], however, this is not the case for lithosphere thicker than 150 km and for Archean lithosphere from ref. [46] that cover smaller areas (Fig 1A).

### 2.3. Reconstructions of past global mantle flow

We reconstruct the evolution of mantle flow from one billion years ago to the present using *CitcomS* [49] in which the mantle is a shell represented with finite-elements in spherical geometry. We use $129 \times 129 \times 65 \times 12 \approx 13$ million elements to obtain an average resolution of ~50 km $\times$ 50 km $\times$ 15 km at the surface, ~40 km $\times$ 40 km $\times$ 100 km in the mid-mantle, and ~28 km $\times$ 28 km $\times$ 27 km at the core-mantle boundary (CMB). We use a version of *CitcomS* [50] modified to progressively assimilate the thermal structure of the lithosphere and of subducting slabs (to ~350 km depth) determined from the synthetic age of the ocean floor [51] as well as surface velocities obtained from tectonic reconstructions with continuously closing plate polygons [52]. These boundary conditions are assimilated in one-million-year intervals. This

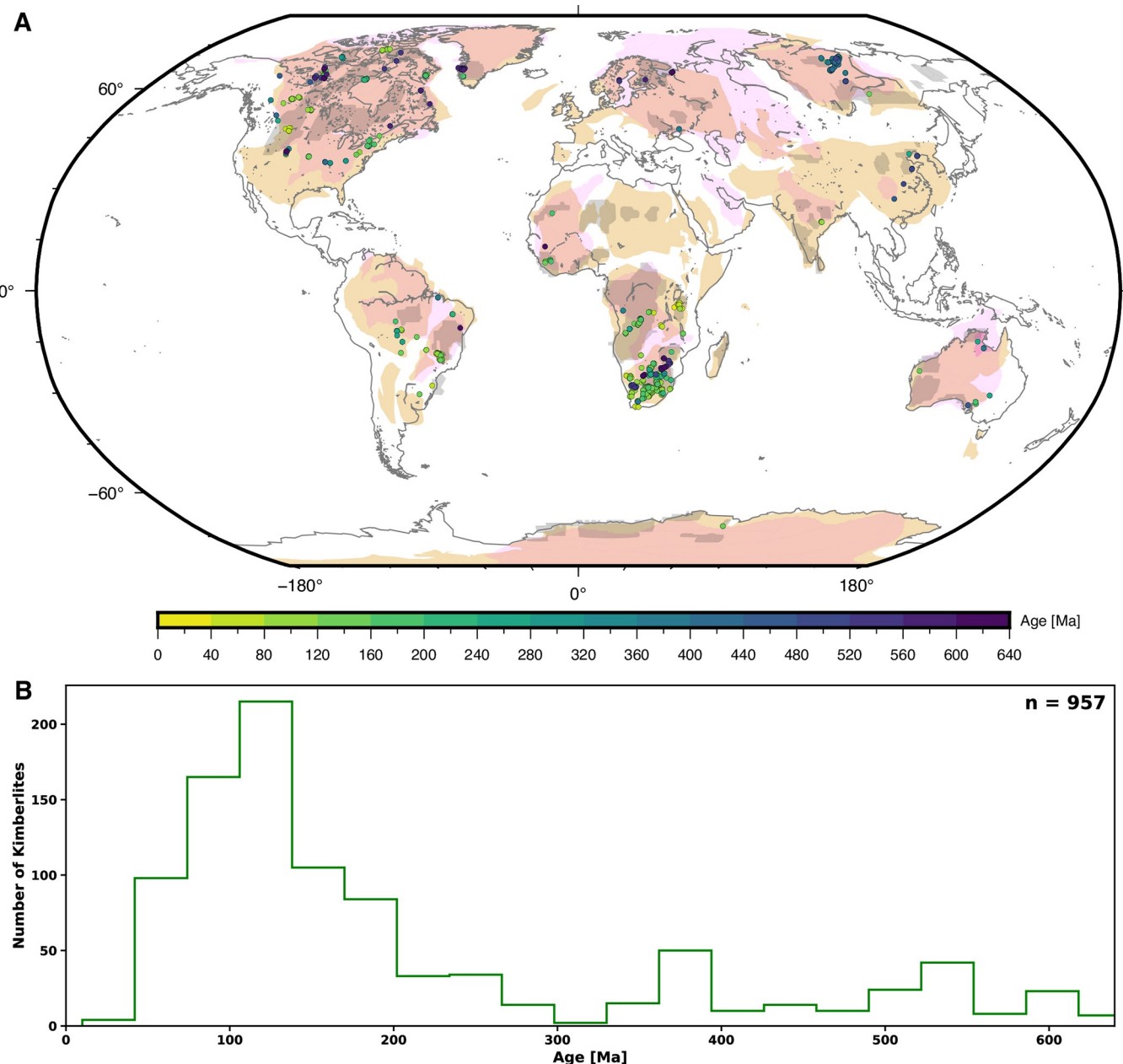

**Fig 1. Distribution of thick or ancient lithosphere and kimberlite eruptions. A/** 957 kimberlite eruptions from ref. [22] from 640 Ma to the present, sampled in 20 Myr increments and coloured by age, and outlines of thick or ancient lithosphere considered in this study: magenta polygons show lithosphere thicker than 150 km in all four lithospheric thickness models considered in ref. [5], beige polygons are tectonic blocks from ref. [38] and grey polygons are blocks of tectonothermal age greater than 2.5 Ga from ref. [46]. Coastlines are shown in grey. This figure was created with GMT6 [47] and the Global Self-consistent, Hierarchical, High-resolution Geography Database (GSHHG) coastlines are republished from [48] under a CC BY license, with permission from Paul Wessel, original copyright 1996. **B/** Histogram showing the temporal distributions of kimberlites occurrences from ref. [22] from 640 Ma. None of the data in this figure are proprietary.

approach makes it possible to obtain one-sided subduction in time-dependent global mantle convection models with computationally affordable resolution and viscosity variations.

We use the extended-Boussinesq approximation, which accounts for viscous dissipation, an adiabatic temperature gradient, internal heating and a decrease in the coefficient of thermal expansion by a factor of two over the thickness of the mantle [53]. Convection vigour was

**Table 1. Parameters varied across mantle flow models.**

| Case number | Reconstruction | Basal layer viscosity pre-factor | Initial slab depth (km) | Basal layer density $\delta\rho_b$ (%) |
|---|---|---|---|---|
| C1 | M21NNR | 10 | 1,000 | +1.02 |
| C2 | M21NNR | 10 | 1,000 | **+1.43** |
| C3 | M21NNR | **1** | **550** | **+2.05** |
| C4 | M21NNR | **1** | **550** | +1.02 |
| C5 | M21NNR | 10 | **550** | +1.02 |
| C6 | **M21** | 10 | 1,000 | +1.02 |

Values differing from Case 1 are in bold.

determined by the Rayleigh number $Ra = (\alpha_0 \rho_0 g_0 \Delta T h_M^3)/(\kappa_0 \eta_0)$, where the subscript "0" indicates reference values, with $\alpha_0$ the coefficient of thermal expansion, $\rho_0$ = 4,000 kg m$^{-3}$ is the density, $g_0$ = 9.81 m s$^{-2}$ the gravitational acceleration, $\Delta T$ = 3,100 K the temperature change across the mantle, $h_M$ = 2,867 km the thickness of the mantle, $\kappa_0$ = 1 × 10$^{-6}$ m$^2$ s$^{-1}$ the thermal diffusivity and $\eta_0$ = 1.1 × 10$^{21}$ Pa s is the viscosity. With the values listed above $Ra$ = 7.8 × 10$^7$. The dissipation number that controls shear heating is $Di = (\alpha_0 g_0 R_0)/C_{p_0}$, and with $R_0$ = 6,371 km being the radius of Earth and $C_{p_0}$ = 1200 J kg$^{-1}$ K$^{-1}$ the reference heat capacity we obtain $Di$ = 1.56. Viscosity varies with depth, composition, temperature and pressure following

$$\eta = \eta(r)\,\eta_0\,\eta_C\,\exp\left\{\frac{[E_\eta + \rho_0 g Z_\eta (R_0 - r)]}{[R(T + T_{off})]} - \frac{[E_\eta + Z_\eta (R_0 - R_c)]}{[R(T_{CMB} + T_{off})]}\right\},$$ with $\eta(r)$ = 0.02 above 160 km depth and

between 310–660 km depth, $\eta(r)$ = 0.002 between 160–310 km depth (asthenosphere), and $\eta$($r$) = 0.02 below 670 km depth (lower mantle). The compositional viscosity pre-factor $\eta_C$ was varied across model cases for the basal layer (see Table 1). $r$ is the radius, $R_C$ = 3,504 km is the radius of the core, $E_\eta$ = 275 kJ mol$^{-1}$ is the activation energy, $Z_\eta$ = 2.1 × 10$^{-6}$ m$^3$ mol$^{-1}$ is the activation volume, $R$ = 8.31 J mol$^{-1}$ K$^{-1}$ is the universal gas constant, $T$ is the dimensional temperature, $T_{off}$ = 452 K is a temperature offset and $T_{CMB}$ = 3380 K is the temperature at the core-mantle boundary. The viscosity pre-factor, activation energy, activation volume and temperature offset were selected to obtain variations in viscosity over three orders of magnitude (viscosity variations were limited to the range 1.1 × 10$^{20}$ Pa s—2.2 × 10$^{23}$ Pa s) across the range of temperatures and pressures (Fig 2).

The initial condition at 1 Ga consists of an adiabatic temperature profile between two thermal boundary layers, with slabs inserted down to either 550 km or 1,000 km depth (Table 1) with a dip of 45° down to 425 km and a dip of 90° below 425 km depth. For cases with slabs inserted down to 1,000 km depth, slabs are twice as thick in the lower mantle than in the upper mantle to account for advective thickening in the more viscous lower mantle [55]. The basal thermal boundary layer is initially 225 km thick and includes a 113-km thick layer [2% of the volume of the mantle; 56] that is denser than ambient mantle. The density contrast between the basal material and ambient mantle, obtained using tracers, was varied between +1.02% and +2.05% across model cases (Table 1).

As slabs sink into the mantle, the initially homogenous basal thermochemical layer is deformed [36, 37] and forms basal mantle structures of changing area. To assess this change, we quantify the evolution of the fractional area ($f_a$) of high temperature clusters in the models; we also compare it to the area of slow-velocity clusters in tomographic models.

## 2.4. Imaging lower mantle structures through cluster analysis

We consider six global S-wave tomographic models that use variations in seismic wave velocities to map mantle structure with respect to a reference model such as PREM [57]: SAW24B16

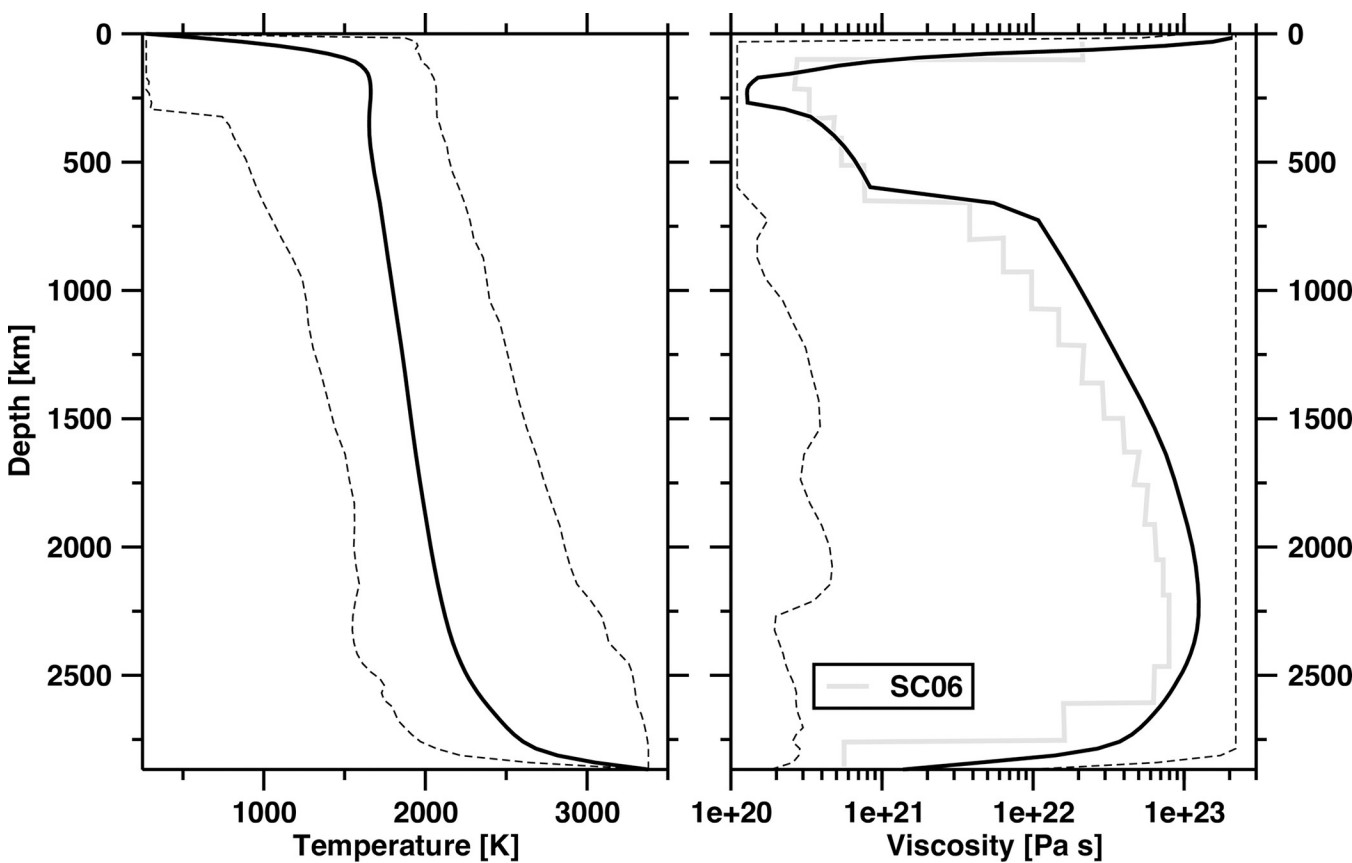

**Fig 2. Temperature and viscosity with depth.** Horizontally averaged present-day mantle temperature (left-hand side) and viscosity (right-hand side) for Case 2. Solid lines show the average, and dashed lines show the minimum and maximum. The grey line on the right-hand side panel is a viscosity profile adjusted to fit the geoid and post-glacial rebound [54].

[58], S362ANI [59], GyPSuM-S [60], S40RTS [61], Savani [62] and SEMUCB-WM1 [63]. As in ref. [64], the structure of the lower mantle is represented by cluster analysis of seismic wave velocities in tomographic models (only for the present-day) and mantle temperature in mantle flow models (present-day and back in time in 20 Myr increments), respectively. We use *k*-means clustering [65], a procedure that minimizes the variance in squared Euclidean distance between vectors to separate ~200,000 points equally-spaced on Earth's surface into two clusters with similar variations between 1,000–2,800 km depths in either seismic velocity (for tomographic models, Fig 3A and 3C) or mantle temperature (for mantle flow models, Fig 3B and 3D).

## 2.5. Quantifying model success

**2.5.1. Success for the present-day: Match with tomographic models.** We assess the success of mantle flow models in reproducing the present-day structure of the mantle by quantifying the match between cluster maps for tomographic models and flow models as in ref. [64]. The accuracy $Acc = (TP+TN)/A$ involves the area of true positive (*TP*, orange in Fig 4) match (where high-temperature clusters from a given flow model intersect slow-velocity clusters from a given tomographic model), and the area of true negative (*TN*, grey in Fig 4) match (where low-temperature clusters from a given flow model intersect fast-velocity clusters from a given tomographic model) over the total area *A*. The sensitivity (sometimes called recall or

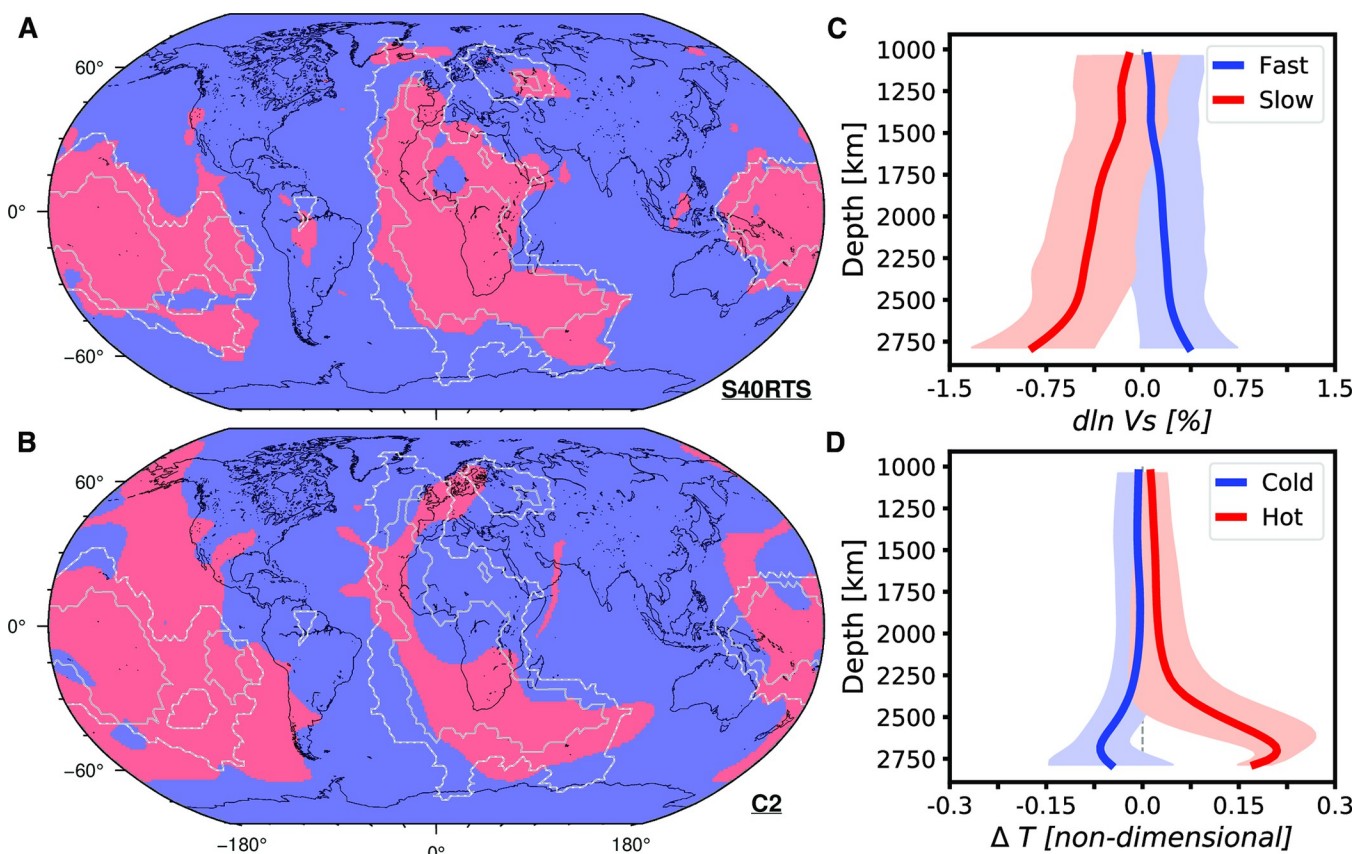

**Fig 3. Cluster analysis of tomographic and mantle flow models. A, B/** spatial distribution of two lower mantle regions revealed by k-means cluster analysis between 1,000 km and 2,800 km depths for **A/** tomographic model S40RTS [61] and **B/** mantle flow model Case 2. Coastlines are shown in black, and the grey lines indicate a value of five (solid) and a value of one (dashed) in a vote map of low-velocity regions from five S-wave tomographic models [66]. These figure panels were created with GMT6 [47] and the Global Self-consistent, Hierarchical, High-resolution Geography Database (GSHHG) coastlines are republished from [48] under a CC BY license, with permission from Paul Wessel, original copyright 1996. **C, D/** depth profiles of clustered properties in each cluster: **C/** velocity profiles in high-velocity (fast, in blue) and low-velocity (slow, in red) regions for S40RTS and **D/** temperature profiles in low-temperature (cold, in blue) and high-temperature (hot, in red) regions for Case 2. The solid curves are the mean, and the transparent envelopes are the associated standard deviation. None of the data in this figure are proprietary.

true positive rate) $S = TP/(TP+FN)$ involves the true positive area (*TP*, orange in Fig 4) and the false negative (*FN*, blue in Fig 4) area (where low-temperature clusters from a given flow model intersect slow-velocity clusters from a given tomographic model).

**2.5.2. Success and predictions in the past: Distance between reconstructed kimberlite eruptions and model high kimberlite potential areas.** We reconstruct the three considered sets of thick or ancient lithosphere back in time and compute the intersections of these polygons with either present-day LLSVPs (assumed to be stationary and rigid back in time) mapped from cluster-analysis of the six considered S-wave tomographic models, or time-dependent basal mantle structures mapped with cluster-analysis in the seven considered flow models. These intersections are considered regions where kimberlites are likely to occur (high kimberlite potential regions). We compute time-dependent potential maps as the distance from these intersections in the mantle reference frame, as well as relative potential maps that summarise potential over a given period in the plate frame of reference; in the latter, potential is obtained by summing intersections between thick or ancient lithosphere and basal mantle structures over time, with values of one within intersections (prospective area at a given time)

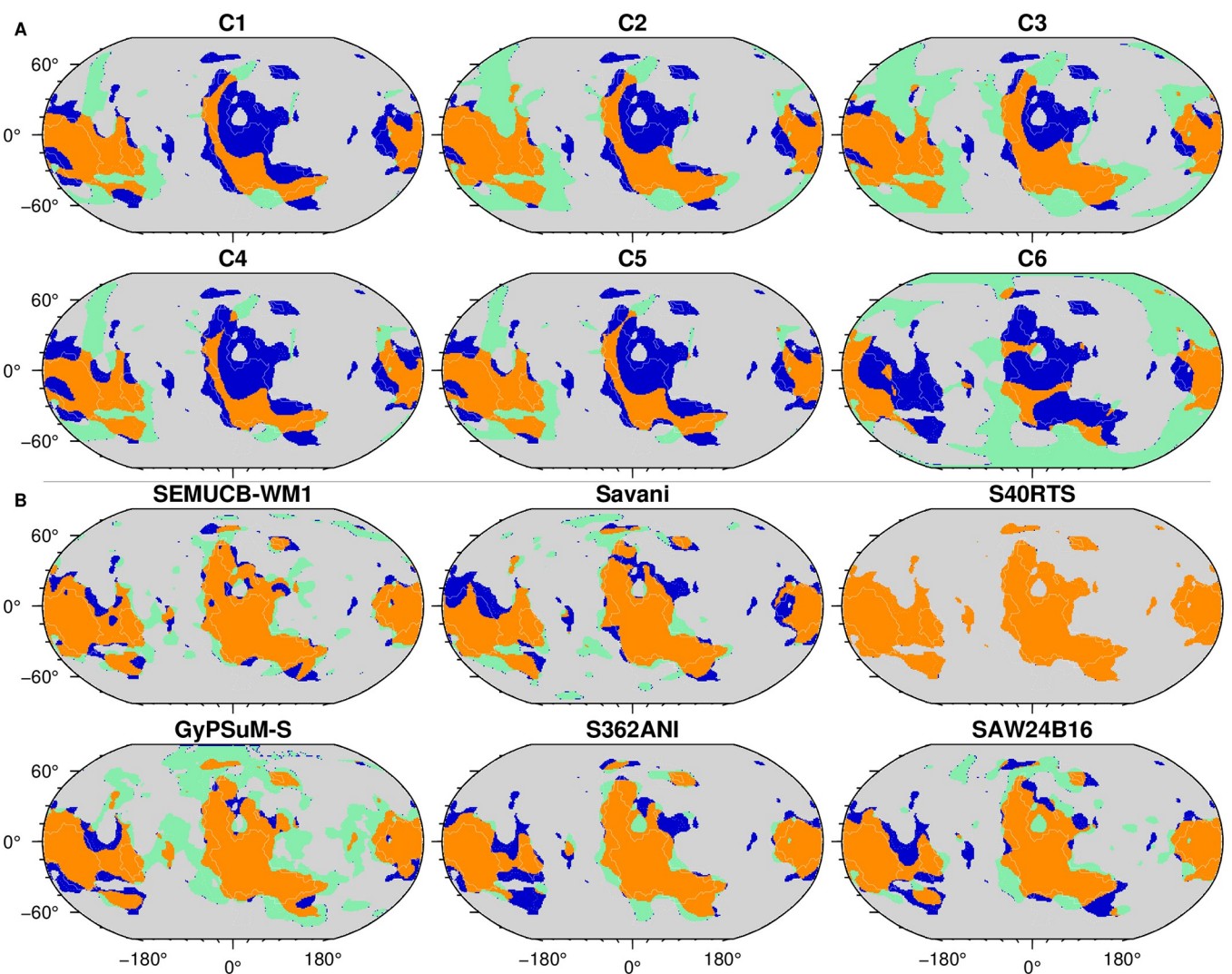

**Fig 4. Spatial match of lower mantle structure between mantle flow and tomographic models with respect to tomographic model S40RTS. A/**
Comparison of present-day mantle structure for S40RTS and for mantle flow model Cases 1–6 as indicated. Orange (true positive) indicates low-velocity tomography and high-temperature regions, grey (true negative) indicates high-velocity tomography and low-temperature regions, green (false positive) indicates high-velocity tomography and high-temperature regions and blue (false negative) indicates low-velocity tomography and low-temperature regions. **B/** Comparison of mantle structure imaged by different tomographic models, using S40RTS as a reference. Orange (true positive) indicates low-velocity regions for both models, grey (true negative) indicates high-velocity regions for both models, green (false positive) indicates high-velocity regions for S40RTS and low-velocity regions for other models and blue (false negative) indicates low-velocity regions for S40RTS and high-velocity regions for other models. This figure was created with GMT6 [47].

and zero outside. "Relative" potential is obtained by normalising the result with respect to the highest.

To evaluate the success of models in predicting past kimberlite eruptions, we measure the distance between each reconstructed kimberlite and the nearest intersection between thick or ancient lithosphere and basal mantle structure. These minimum distances, cumulated over time, are summarised in one empirical distribution function for each combination of thick or ancient lithosphere and basal mantle structures, and the median and average minimum distances from these EDFs are used as indicators of model success.

## 3. Results

### 3.1. Area of basal mantle structures in tomographic and mantle flow models

The fractional area $f_a$ of basal mantle structures affects the match between mantle flow models and tomographic models, as well as the minimum distances between kimberlites and potential areas because models with larger $f_a$ result in larger potential areas, leading to smaller minimum distances to kimberlites.

For LLSVPs mapped as slow-velocity clusters from tomographic models, $f_a$ ranges between 0.31 and 0.39 to the exception of GyPSuM-S for which the area is 0.52 (Fig 5). In the mantle flow models, the fractional area covered by the high-temperature cluster changes from $f_a = 1.0$ in the initial condition until it stabilizes somewhere between 0.2 and 0.5 depending on the density of the basal layer (Fig 5A and Table 1). Over time, sinking slabs reduce the fractional area of the high-temperature cluster. This effect is most pronounced in cases with slabs initially inserted down to 1,000 km depth (Cases 1, 2 and 6 in Fig 5A). In all cases, the area of basal mantle structures stabilises by about 640 Ma (360 Myr into the model run), which is about twice the time it takes for slabs to sink to the deep mantle in these models [67]. For this reason, we analyse models for the last 640 Myr.

In the flow models, the fractional area covered by basal mantle structures $f_a$ also depends on the buoyancy of the basal layer: it is larger for a denser basal layer ($\bar{f}_a \approx 0.47$ for Case 3 with $\delta\rho_b = +2.05\%$) and smaller for a less dense basal layer ($\bar{f}_a \approx 0.26$ for Cases 1, 4 and 5 with $\delta\rho_b = +1.02\%$). Case 2 with $\delta\rho_b = +1.43\%$ results in $\bar{f}_a \approx 0.37$ which is consistent with tomographic models ($0.31 < f_a < 0.39$). For a given density of the basal layer, $f_a$ is larger for M21 ($\bar{f}_a \approx 0.31$ for Case 6) than for M21NNR ($\bar{f}_a \approx 0.26$ for Cases 1, 4 and 5), due to the different location of subduction zones with different reference frames (Fig 4A).

### 3.2. Match between predicted mantle structure and tomographic models

The shape and size of basal thermochemical structures differ between model cases. Flow models that better match present-day tomographic models can be extrapolated back in time with more confidence, because the structure of the lower mantle is shaped by the history of subduction. To evaluate how the mantle structure from flow models matches that from tomographic models, we compute the accuracy $Acc = (TP+TN)/A$ and sensitivity $S = TP/(TP+FN)$ between tomographic models and between each mantle flow model and tomographic model (Fig 6). The accuracy is a global match, whereas the sensitivity is a match between imaged and predicted and slow/hot basal mantle structure.

Overall, the match between tomographic models ($0.58 < S < 0.86$ and $0.76 < Acc < 0.88$, excluding auto-correlation for which both scores are equal to one, Fig 6) is better than the match between mantle flow and tomographic models ($0.33 < S < 0.74$ and $0.54 < Acc < 0.78$). This result is expected since tomographic models are generally similar to one another in the way they were built, whereas the forward mantle flow models that cover one billion years of tectonic history are completely independent from tomographic models. Within tomographic models, the largest average sensitivities and accuracies are obtained for tomographic models S40RTS and SEMUCB-WM1. In contrast, GyPSuM-S appears to be an outlier for which the accuracy and sensitivity are systematically lower than for other tomographic models, because $f_a$ is greater in that tomographic model (Figs 4–6). Amongst mantle flow models, the sensitivity is largest between Case 1, 4 and 5 and S40RTS, Savani and S362ANI; it is also relatively large for Case 2. The accuracy is largest for the same mantle flow models but with S40RTS and SEMUCB-WM1. The accuracy and sensitivity are lowest for Case 6 that is based on

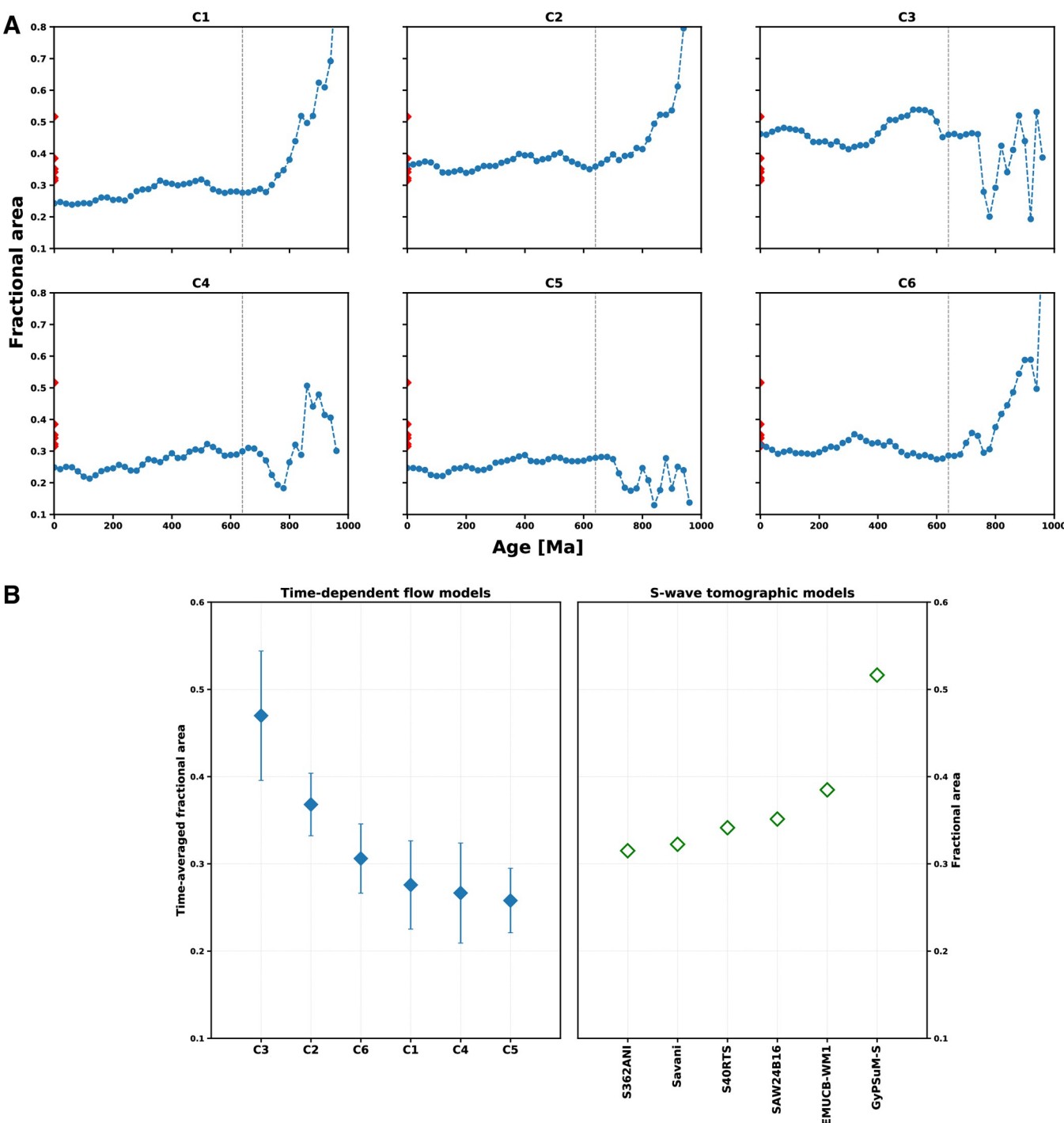

**Fig 5. Fractional area covered by hot/slow basal mantle structures in mantle flow and tomographic models.** **A/** Evolution of the fractional area of basal mantle structures $f_a$ in flow models (blue circles and lines) and for tomographic models (red diamonds at present-day). The dashed vertical line is at 640 Ma. **B/** Fractional area of basal mantle structures averaged from 640 Ma ($\bar{f_a}$, blue diamonds with standard deviation shown as error bars on the left-hand-side panel, sorted by decreasing $\bar{f_a}$), compared to present-day $f_a$ for tomographic models (open green diamonds on the right-hand-side panel, sorted by increasing $f_a$).

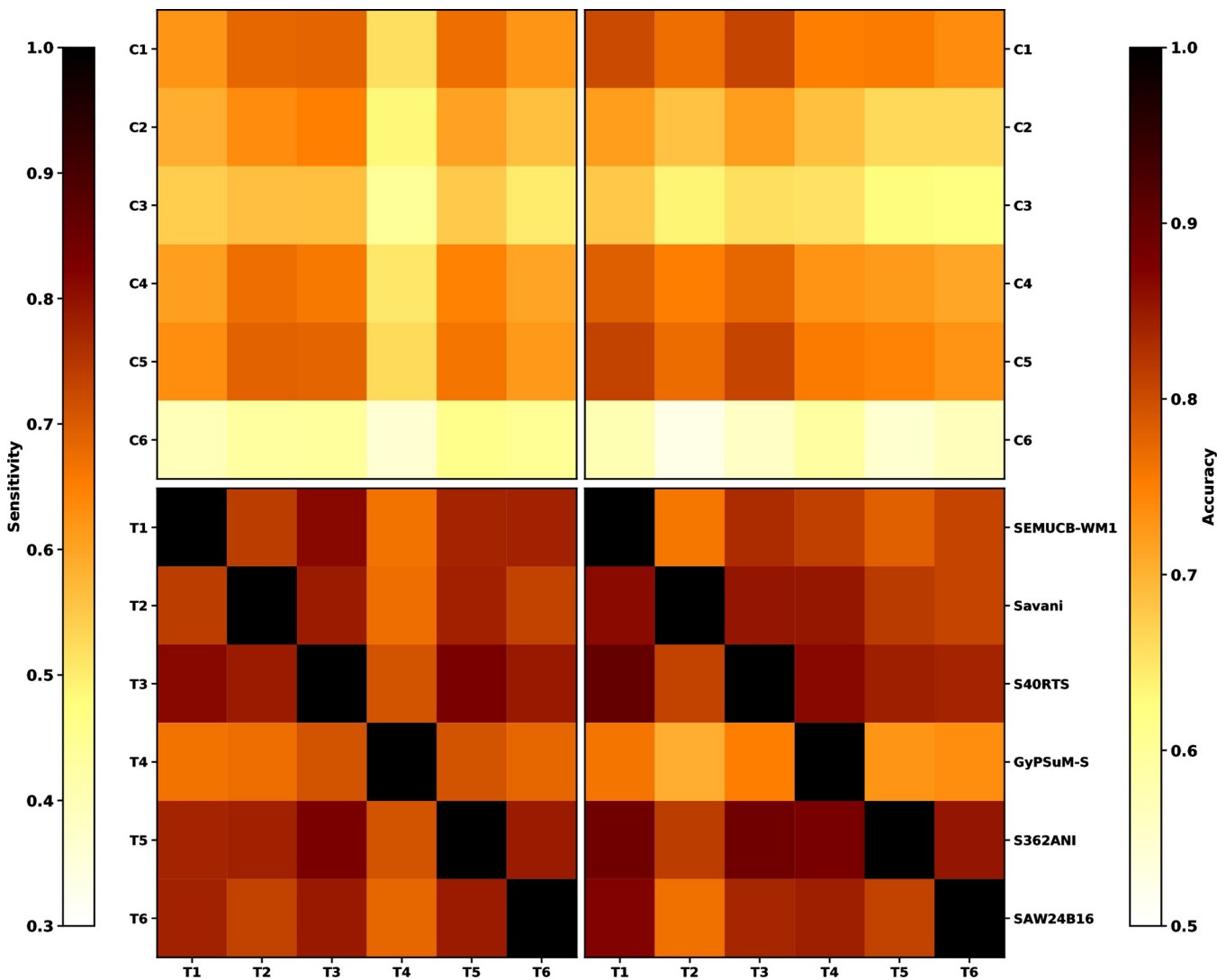

**Fig 6. Quantitative match between predicted and imaged basal mantle structures.** Match between hot basal mantle structures predicted by flow models and LLSVPs in seismic tomography models as mapped between 1,000 km and 2,800 km depths using cluster analysis. Left-hand-side panels show the sensitivity and right-hand-side panels the accuracy. In the top panels, quantities are derived between six flow models and six tomographic models, and in the bottom panels they are derived between six tomographic models.

reconstruction M21, in which net lithospheric rotation present in the surface boundary conditions induces wholesale motion of the lower mantle and basal mantle structures [44]. For this reason, we focus on M21NNR in the following.

There is a trade-off between $f_a$ and $S$ and $Acc$: model cases in which the basal layer is less dense (Cases 1, 4 and 5) result in larger $S$ and $Acc$ (match to tomographic models), however in these cases $f_a$ is smaller than in tomographic models. Case 2 ($\delta\rho_b$ = +1.43% and reconstruction M21NNR), for which $f_a$ is comparable to tomographic models and $Acc$ and $S$ are reasonably large, is our preferred mantle flow model case. In the following, we first illustrate results for Case 2 and tomographic models S40RTS (which is the best tomographic model based on $Acc$ and $S$) before summarising results for all models.

### 3.3. Reconstructing kimberlites, thick or ancient lithosphere and basal mantle structures back to 640 Ma

As an example, we reconstruct kimberlites occurrences from ref. [22] and lithosphere thicker than 150 km, and consider basal mantle structures in S40RTS and Case 2 back to 640 Ma (Fig 7 and S1 Video). S40RTS better captures reconstructed southern African kimberlites than Case 2 back to 80 Ma (S1 Video). However, before 80 Ma, Case 2 better captures the peak of South African kimberlite activity at 100 Ma. All flow models based on reconstruction M21NNR capture the southern African kimberlite blooms back to around 200 Ma, as illustrated by comparing S40RTS and Case 2 at 120 Ma, 160 Ma and 240 Ma (Fig 7). S40RTS and Case 2 are both generally consistent with kimberlite eruptions on the Siberian block back to 360 Ma. S40RTS does not match the kimberlite records in Antarctica at 120 Ma and Australia at 240 Ma, whereas Case 2 does (Fig 7). Case 2 almost captures the kimberlite occurrence in North China at 240 Ma (Fig 7G), although the North China craton is thinner than 150 km following lithospheric delamination [68] and the model basal mantle structure is offset to the East of the North China block from ref. [38] (Fig 7G).

Neither S40RTS nor Case 2 capture kimberlite magmatism in the Americas at 120 Ma. However, Case 2 better captures kimberlite occurrences than S40RTS in southern Africa and Antarctica at that time. S40RTS encompasses the kimberlite occurrence in Western Australia at 120 Ma whereas Case 2 does not (Fig 7A and 7E). In contrast, Case 2 captures the kimberlite occurrence in the Northern Territory of Australia at 240 Ma and 360 Ma, whereas S40RTS does not (Fig 7C and 7G).

At 160 Ma (Fig 7B and 7F), both tomographic and flow models capture some north-eastern Canadian kimberlites although neither models matches the North American kimberlites until 170 Ma; as in previous work [25]. At 240 Ma, S40RTS captures Siberian and north-eastern Canadian kimberlites (Fig 7C), and Case 2 captures kimberlites in Siberia, Africa, and Australia (Fig 7G). Both models capture some Siberian kimberlites at 360 Ma (Fig 7D and 7H).

### 3.4. Global kimberlite potential maps

**3.4.1. Global kimberlite potential maps in the mantle frame of reference.** We next present the time-dependent distance to polygons defined by the intersection of reconstructed lithosphere thicker than 150 km and basal mantle structures as mapped using cluster analysis. In these kimberlite potential maps, the distance is zero within the intersection and increases away from the intersections (Fig 8 and S2 Video).

Kimberlite potential maps for S40RTS and Case 2 both indicate southern African, eastern Canada, Greenland, northern Europe, the eastern African coast, and the west coasts of India and Madagascar as high-potential areas for kimberlite magmatism over the last 240 Myr. Again, the predicted kimberlite potential before 80 Ma in southernmost Africa is smaller for S40RTS than for Case 2 (S2 Video). S40RTS predicts lower kimberlite potential than Case 2 for South America, Australia and Antarctica at 360 Ma (Fig 8D and 8H). Since reconstructed kimberlites are generally within or close to high kimberlite potential areas (Fig 8), we next quantify the distance of reconstructed kimberlites to high kimberlite potential areas to evaluate the success of different models.

**3.4.2. Global kimberlite potential maps in the plate frame of reference.** We next present relative kimberlite potential over a given period in the plate frame of reference and within presently emerged continents (Fig 9A–9D). This shows that for reconstruction M21NNR, the largest predicted kimberlite potential is in central and western Africa for S40RTS (Fig 9A and 9C), whereas it is in southern Africa for Case 2 (Fig 9B and 9D). This result is similar over 320 Myr (Fig 9A and 9B) and 640 Myr (Fig 9C and 9D). Because more regions are at the

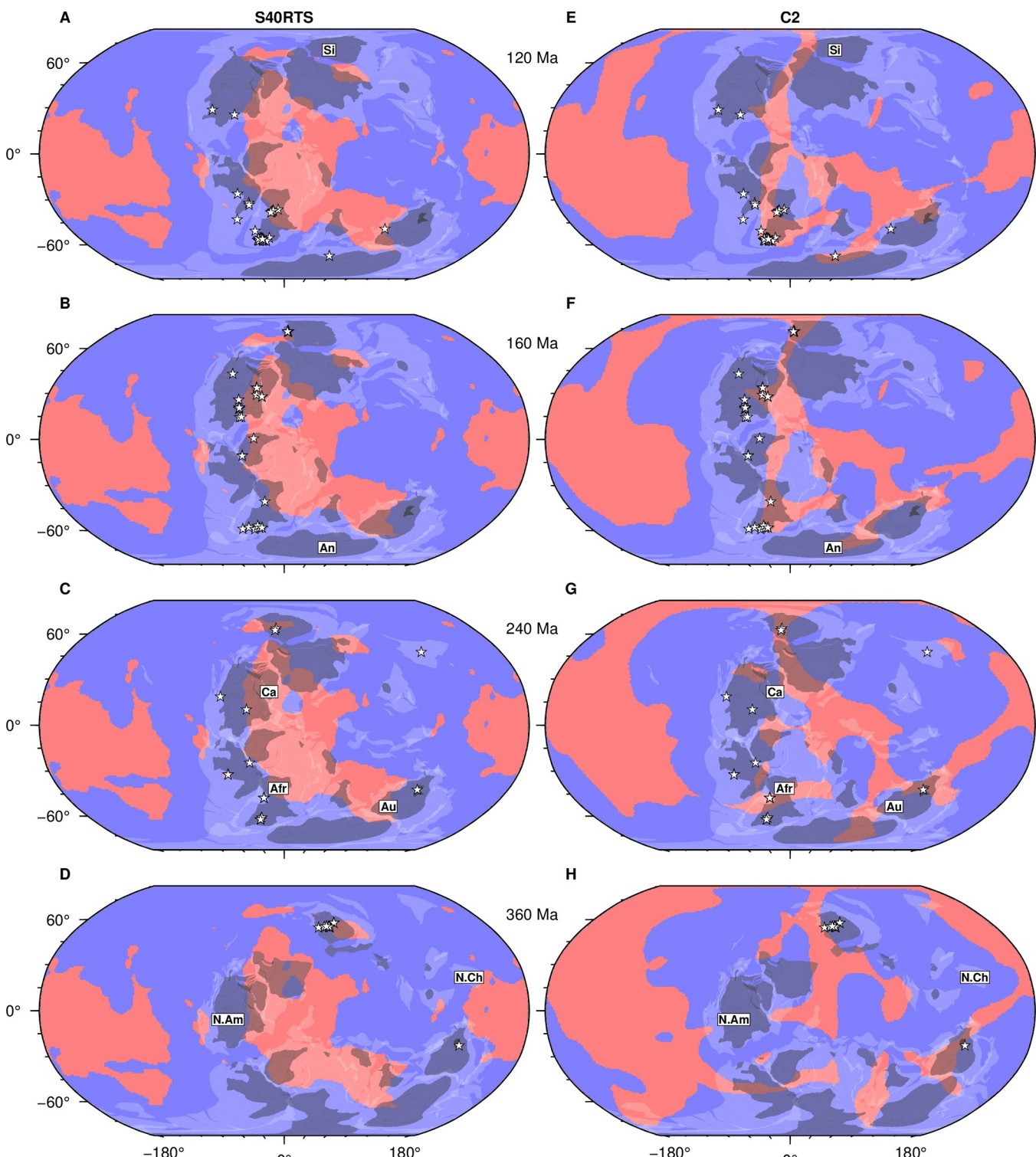

**Fig 7. Evolution of the location of slow/hot basal mantle structures and kimberlite eruptions.** Kimberlite eruption locations (stars) sampled in 20 Myr increments from ref. [22] continent outlines from ref. [38] (white transparent polygons) and lithosphere thicker than 150 km derived from ref. [5] (dark grey transparent polygons), all reconstructed at 120 Ma (**A, E**), 160 Ma (**B, F**), 240 Ma (**C, G**) and 360 Ma (**D, H**) with reconstruction M21NNR. The background shows cluster maps with basal mantle structures in pink for tomographic model S40RTS (**A-D**) and for mantle flow model Case 2 (**E-H**). The intersections of white and pink areas are high-potential regions for kimberlites. Labels: **A**/ "Si": Siberia; **B**/ "An": Antarctica; **C**/ "Afr": Africa, "Au": Australia, "Ca": Canada; **D**/ "N.Am": North America. This figure was created with GMT6 [47] and the Global Self-consistent, Hierarchical, High-resolution Geography Database (GSHHG) coastlines are republished from [48] under a CC BY license, with permission from Paul Wessel, original copyright 1996.

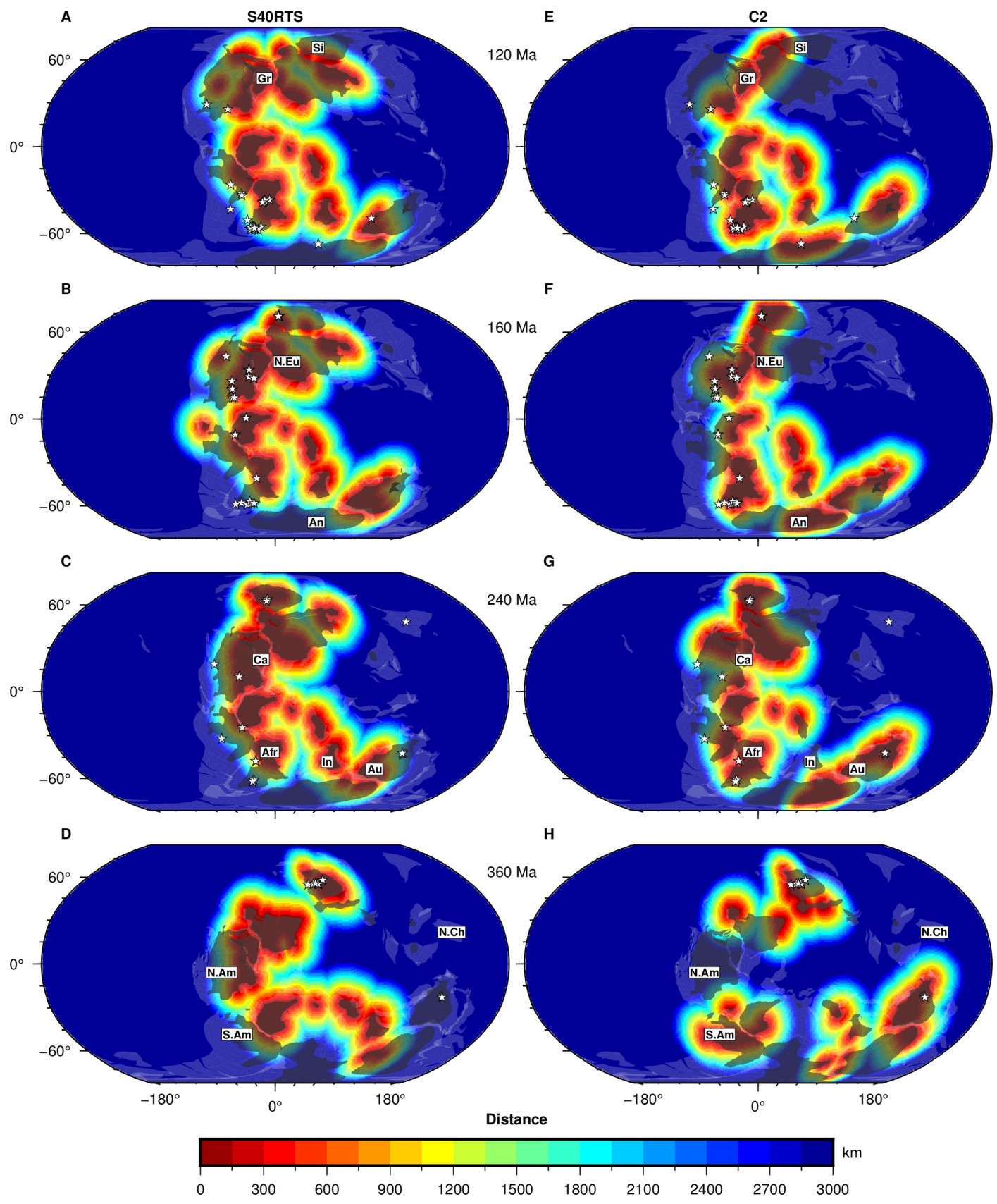

**Fig 8. Global kimberlite potential maps and kimberlite eruptions through time.** Distance to the intersection between basal mantle structures and reconstructed lithosphere thicker than 150 km [5] at 120 Ma (**A, E**), 160 Ma (**B, F**), 240 Ma (**C, G**) and 360 Ma (**D, F**) for tomographic model S40RTS (**A-D**) and mantle flow model Case 2 (**E-H**). Continent outlines from ref. [38] are shown as transparent white polygons, and kimberlite eruptions sampled in 20 Myr increments from ref. [22] are shown as stars; both are reconstructed using M21NNR. Labels: **A/** "Gr": Greenland, "Si": Siberia; **B/** "An": Antarctica, "N.Eu": Northern Europe; **C/** "Afr": Africa, "Au": Australia, "Ca": Canada, "In": India; **D/** "N.Am": North America, "N.Ch": North China, "S.Am": South America. This figure was created with GMT6 [47] and the Global Self-consistent, Hierarchical, High-resolution Geography Database (GSHHG) coastlines are republished from [48] under a CC BY license, with permission from Paul Wessel, original copyright 1996.

intersection between thick lithosphere and basal mantle structures when a longer model duration is considered, the area of high-potential regions is larger over 640 Myr (Fig 9C and 9D) than over 320 Myr (Fig 9A and 9B). Overall, the relative kimberlite potential at kimberlite locations is greater for Case 2 than for S40RTS (Fig 9E and 9F): the peak in the number of kimberlites is for a relative kimberlite potential of about 0.3 for S40RTS (median 0.21 and mean 0.19) and about 0.55 for Case 2 (median 0.24 and mean 0.35) over 320 Ma (Fig 9E); over 640 Ma (Fig 9F) the peak is at ~0.15 for S40RTS (median 0.16 and mean 0.18) and ~0.55 and ~0.75 for Case 2 (median 0.22 and mean 0.33). We note that there are also significant peaks at low-potential (<0.15), and that these peaks are higher for Case 2 than for S40RTS (Fig 9E and 9F).

## 3.5. Distance between kimberlites and predicted high kimberlite potential areas

**3.5.1. Empirical distribution functions.** We sample the distance to the closest high kimberlite potential region (referred to as minimum distance) at each reconstructed kimberlite locations (Fig 8) and report results cumulated over time as empirical distribution functions (EDFs) for each combination of basal mantle structures and thick or ancient lithosphere and for the last 640 Ma (Fig 10). These EDFs show the likelihood that a given proportion of kimberlites falls within a given minimum distance from a high kimberlite potential region. Thus, EDFs with smaller minimum distances reflect cases for which reconstructed kimberlites are closer to predicted high kimberlite potential areas, which indicates more successful models that better account for the kimberlite record. Models with a greater likelihood that distances are equal to zero are also more successful. As expected, distances are smallest for tomographic model GyPSuM-S and flow model Case 3 (Fig 10) because these models result in larger basal mantle structures (Figs 4 and 5).

**3.5.2. Metrics from empirical distribution functions.** To summarise EDFs, we report the mean minimum distance and associated standard deviation for each distribution, as well as the median minimum distance (distance for which the likelihood is equal to 0.5; Fig 11). The mean and median minimum distances confirm a link between reconstructed kimberlite eruption locations, reconstructed thick or ancient lithosphere, and basal mantle structure locations: median minimum distances indicate that 50% of kimberlites are within ~800 km of high kimberlite potential areas for all models (Fig 11A and 11C) and mean minimum distances are less than 1500 km (Fig 11B and 11D). Mean and median minimum distances are similar for mantle flow models and tomographic models (Fig 11), suggesting that mobile and deforming basal mantle structures are as consistent with the kimberlite record as stationary and rigid ones. This first order result, as well as trends between models, are similar over 320 Ma (Fig 11A and 11B) and over 640 Ma (Fig 11C and 11D). Distances are largest (least favourable) for Cases 4 and 5, in which slabs are initially inserted down to 550 km depth and the density of the basal layer is intermediate (+1.02%; Table 1). Standard deviations are large for tomographic model S362ANI and flow model Cases 1, 4 and 5 (Fig 11), reflecting more spread distributions (Fig 10) due to smaller $f_a$ (Fig 4B). In contrast, mean and median distances are smallest for tomographic model GyPSuM-S and mantle flow Case 3, however these two cases do not match

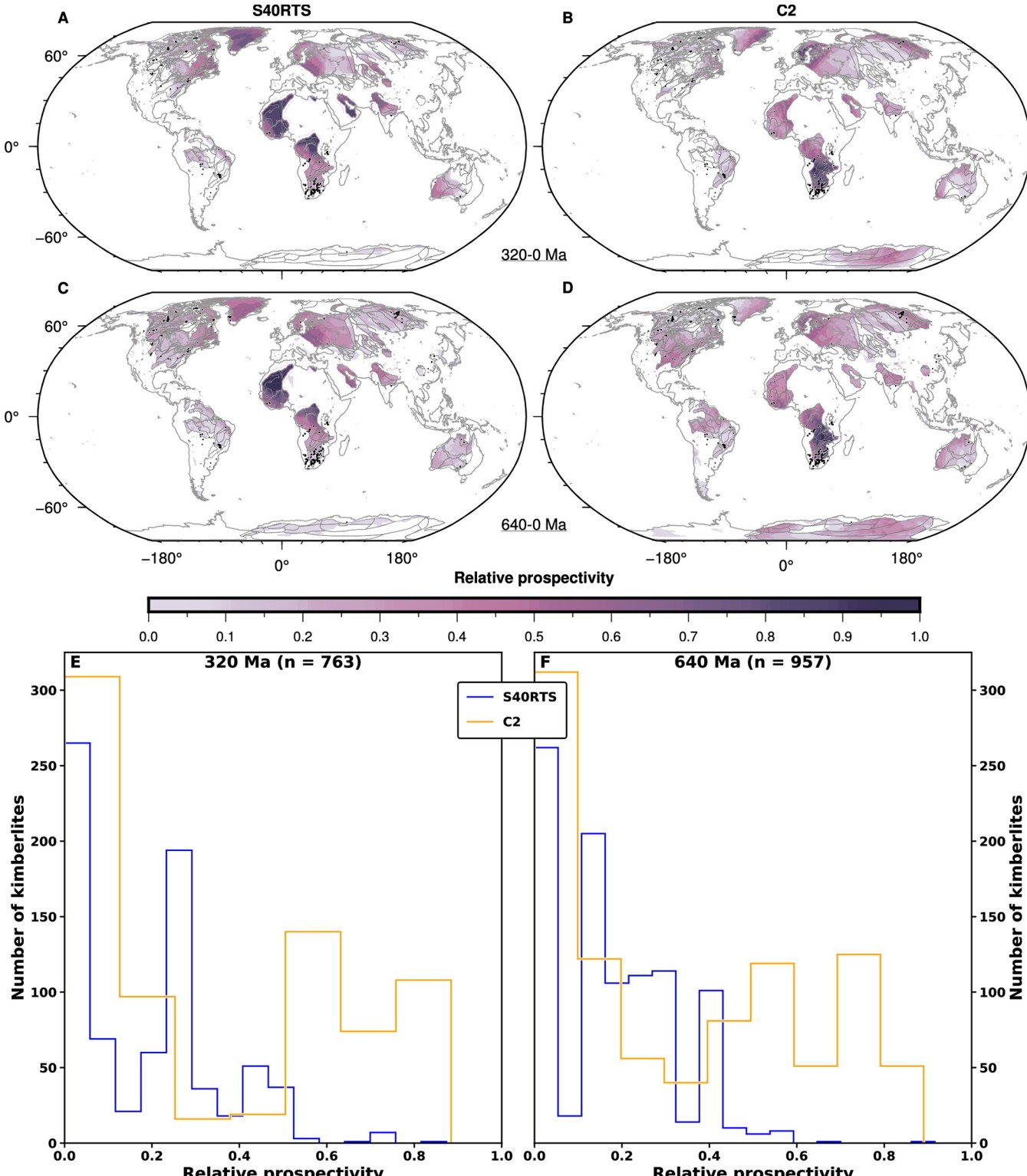

**Fig 9. Predicted relative kimberlite potential. A-D/** Global relative kimberlite potential maps for lithosphere thicker than 150 km from 320 Ma (**A-B**) and from 640 Ma (**C-D**) for tomographic model S40RTS (**A, C**) and Case 2 (**B, D**). Kimberlites from ref. [22] are shown as black circles for the last 320 Ma (**A-B**) and 640 Ma (**C-D**). These figure panels were created with GMT6 [47] and the Global Self-consistent, Hierarchical, High-resolution Geography Database (GSHHG) coastlines are republished from [48] under a CC BY license, with permission from Paul Wessel, original copyright 1996. **E-F/** Distribution of relative kimberlite potential at kimberlite locations shown in **A-D** for S40RTS and Case 2 from 320 Ma (**E**) and from 640 Ma (**F**).

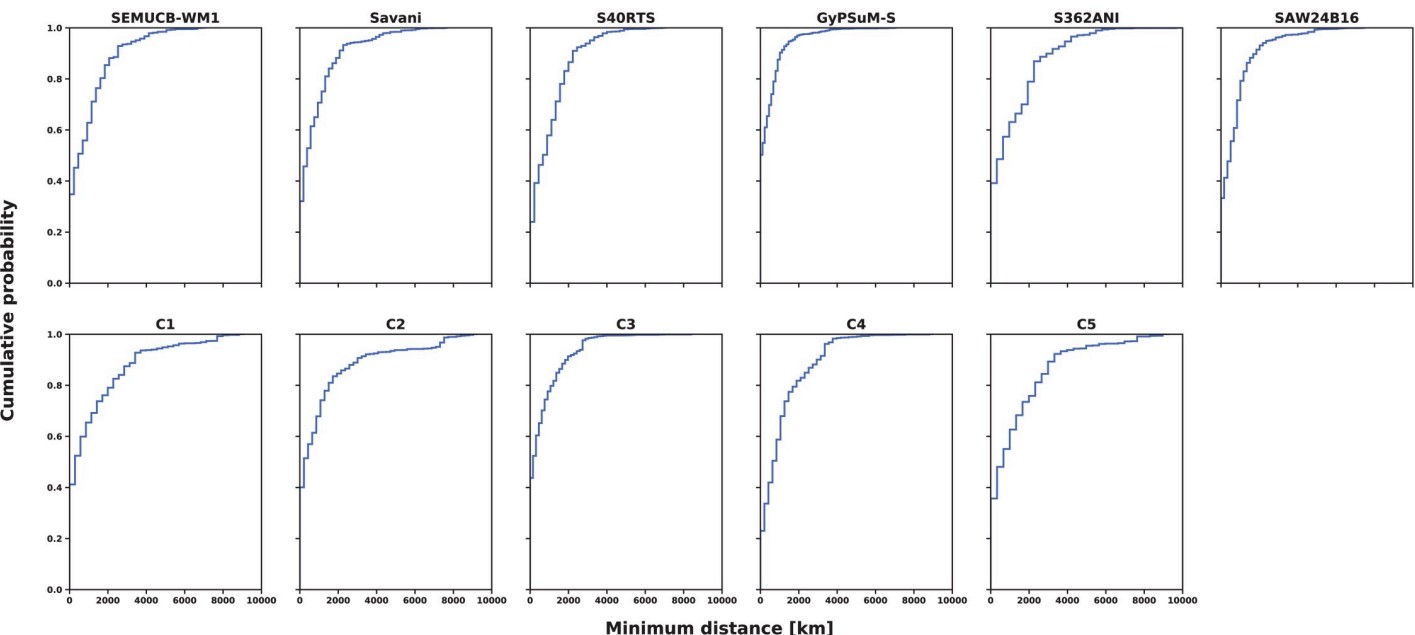

**Fig 10. Distances between kimberlite eruption locations and high kimberlite potential areas.** Empirical Distribution Functions (EDFs) of cumulative minimum distances from 640 Ma between reconstructed kimberlites (sampled in 20 Myr increments) and predicted high kimberlite potential areas for lithosphere thicker than 150 km [5] reconstructed using M21NNR, for all six tomographic models, and mantle flow models Cases 1–5 (that are based on M21NNR).

other tomographic models (Fig 6) due to large basal mantle structures (Fig 5B). Interestingly, mantle flow model Case 2 results in relatively small minimum distances (Fig 11) as well as being compatible with tomographic models (Figs 5 and 6). For Case 2, 50% of kimberlites are within ~250 km of high kimberlite potential areas over 320 Ma (Fig 11A), and 50% of kimberlites are within ~350 km of high kimberlite potential areas over 640 Ma (Fig 11C). When different lithospheric extents are used, trends are similar, and mean and median distances are larger when using the smaller blocks of Archean tectonothermal age from ref. [46], and smaller when using the larger tectonic blocks are used from ref. [38] (Fig 12). These trends are expected, and we note that the difference varies between cases, because the distance depends on the location of basal mantle structures as well as the location of reconstructed thick or ancient lithosphere (Figs 7 and 8, S3 and S4 Videos). When reconstruction M21 (in which the net rotation of the lithosphere is not removed) is used, mean and median distances are approximately twice as large than with reconstruction M21NNR for tomographic models (S1 Fig).

## 4. Discussion

### 4.1. Model predictions: Regions of potential interest for large-scale kimberlite exploration

The global kimberlite potential maps presented in this study (Figs 7 and 8, S1–S4 Videos) may be used for global-scale kimberlite exploration, placing them at the base of the hierarchy of scale-dependent targeting processes [69]. Thick or ancient lithosphere that have repeated intersections with model slow or hot clusters are more likely to have undergone kimberlite magmatism and are therefore of potential interest for large-scale kimberlite exploration. Kimberlite potential maps for S40RTS and Case 2 capture regions of prominent kimberlite magmatism such as Africa [70] and Siberia [71] (Figs 8 and 9D). Case 2 also predicts some kimberlite activity in northwest Australia, Greenland, northeast Canada and Northern Europe (Figs 8 and 9D).

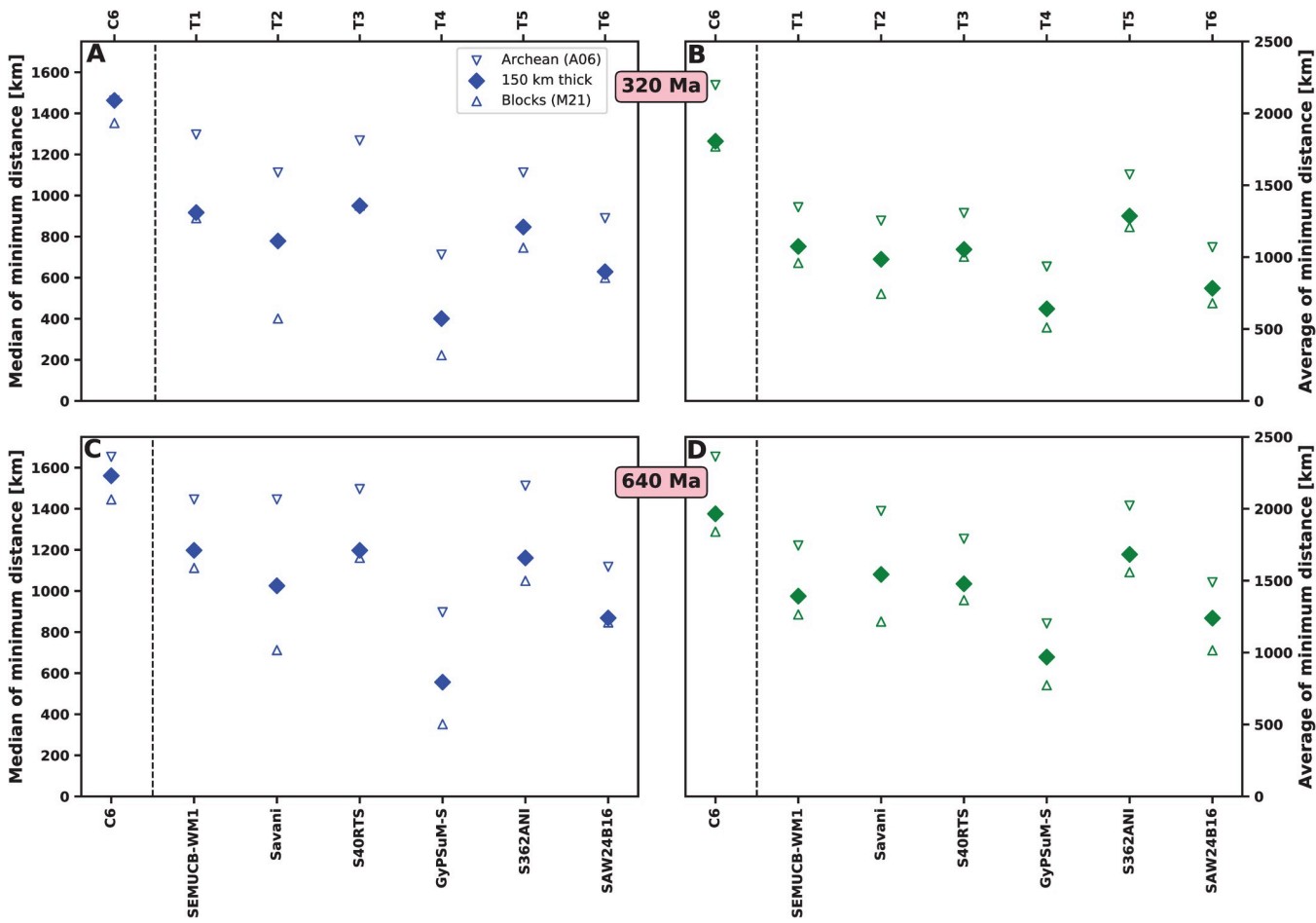

**Fig 11. Results for lithosphere thicker than 150 km [5] and reconstruction M21NNR.** Metrics from EDFs for lithosphere thicker than 150 km [5] and reconstruction M21NNR for mantle flow and tomographic models from 320 Ma (**A-B**) and from 640 Ma (**C-D**). **A, C/** Median and **B, D/** average minimum distances with one standard deviation as error bar. In each panel, mantle flow model cases are to the left and tomographic models to the right of the dashed vertical line.

Interestingly, Case 2 indicates Antarctica as a location of kimberlite potential for the last 640 Ma (Fig 9D), although there is only one known instance of kimberlite magmatism on the continent [72].

## 4.2. Mobility of basal mantle structures

Previous work has been shown that the reconstructed location of ~1,100 kimberlites over the last 320 Myr are within ~1,650 km of the edge of the African LLSVP as imaged by SMEAN at 2,800 km depth [25]. Interestingly, ~240 kimberlite occurrences from the Slave Province of Canada were more distant from the edge of the African LLSVP. Nevertheless, reconstructed locations of 25 LIPs erupted in the last 297 Myr were also close to the edge of LLSVPs, leading to the hypotheses that 1/ mantle plumes arise from LLSVPs, and 2/ that LLSVPs could have been stationary over the last 300 Myr, and possibly longer [25, 73, 74].

Here, we assumed that a relationship between basal mantle structures and kimberlites exists, and we considered two main scenarios: one in which lower mantle structures are mobile (represented by mantle flow models) and one in which lower mantle structures are stationary over time (represented by tomographic models). Overall, we found that minimum distances

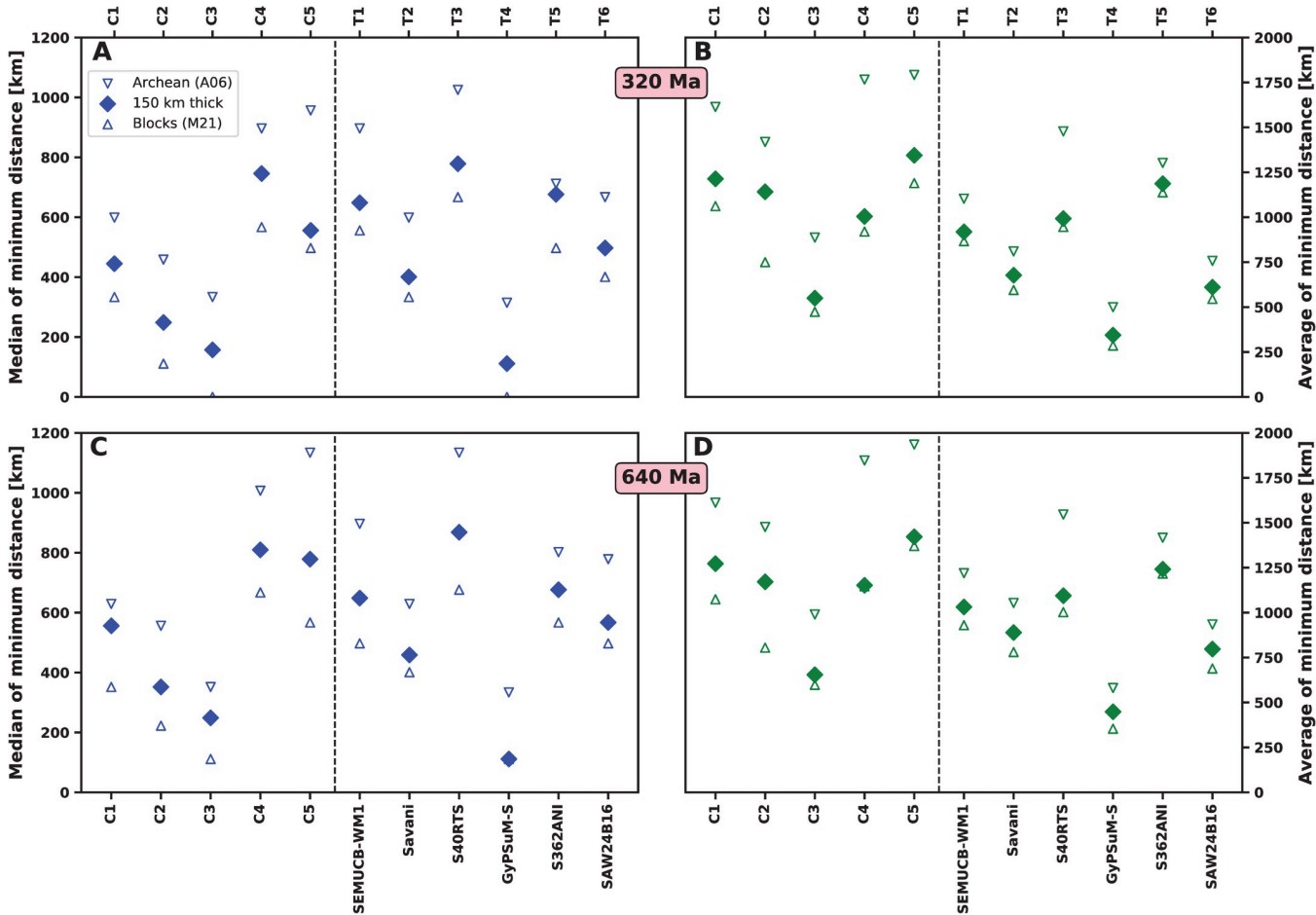

**Fig 12. Results for different lithospheric extents and reconstruction M21NNR.** Same as Fig 11 but for tectonothermal ages greater than 2.5 Ga from ref. [46] (open downward pointing triangles, "Archean (A06)"), lithosphere thicker than 150 km [5] (diamonds, "150 km thick") and tectonic blocks from ref. [38] (open upward pointing triangles, "Blocks (M21)"), using reconstruction M21NNR.

between kimberlites and high-potential areas for kimberlites are similar for the two scenarios (Figs 11 and 12 and S1 Fig), and that mobile basal structures better predict the abundance of kimberlites in southern Africa than stationary basal mantle structures (Figs 7–9). This suggests that kimberlites are as close to mobile basal structures as they are to stationary basal mantle structures, and that there is no need to hypothesise that basal mantle structures have been stationary over time, which is consistent with the mobility of tectonic plates, subduction zones [38, 42] and mantle plumes [75, 76] over time.

### 4.3. Limitations

Our global kimberlite potential maps are based on the assumption that kimberlites are associated with deep mantle upwelling above basal mantle structures. We did not consider shallower controls such as fractures and faults cutting through the lithosphere [21]. In addition, not all kimberlites are necessarily linked to the deep mantle, and kimberlite eruptions may also occur due to partial melting at the base of the lithosphere or above subducting slabs [12, 77] and can be emplaced through weak zones within the lithosphere or close to plate boundaries [21]. Indeed, the lithospheric mantle might be the source of some South African orangeites (formerly known as group II kimberlites) [78], and kimberlites from the Slave Province that are

not closely associated with the African LLSVP (Figs 7–9) could be associated to the history of subduction under North America [79]. Recent geochemical fingerprinting of kimberlites revealed that some kimberlite erupted above basal mantle structures could be derived from a primitive, depleted mantle source, whereas kimberlite erupted away from basal mantle structures could be derived from a source that contains deeply subducted crustal material [80]. Even though upwelling of hot mantle from the deep Earth may provide the source of heat leading to kimberlite eruption, the physical processes that could link basal mantle structures [>2000 km deep; 26] and kimberlite eruption from ~120–300 km depth [12, 13] are not yet firmly established.

Our global kimberlite potential maps also rely on a definition of relevant lithospheric extent. While diamondiferous kimberlites are generally associated with cratons [e.g., 12], the definition of craton may not be limited to the age of the overlying crust [46] and the geochemical composition of the subcontinental lithospheric mantle [45], and a recent definition suggests that lithosphere 150 km thick could be an appropriate proxy for craton extent [4]. We follow this definition here, which results in larger blocks than using tectonothermal ages [46] (Fig 1), leading to smaller distances between kimberlite eruptions and kimberlite potential maps (Fig 12). We derived lithosphere thicker than 150 km from published lithospheric thickness models [5] and we note that this simple proxy does not consider modified cratons [4], most notably the North China Craton that has lost its lithospheric root [68] and contains kimberlites (Fig 1).

The presented reconstructions of past mantle flow do not predict the location of kimberlite eruptions. Modelling kimberlite eruptions requires a horizontal resolution of a few hundred meters, modelling two-phase flow (melt and solid) and considering viscosity contrast over 26 orders of magnitude between the melt with viscosity 0.01 Pa s [81] and the lithosphere with viscosity about 1 x $10^{24}$ Pa s. The presented time-dependent global mantle flow models achieve a horizontal resolution of 50 km, viscosity contrast over three orders of magnitude, and do not consider melting. Achieving kilometre-scale resolution is possible in instantaneous global mantle flow models [82] but remains prohibitively expensive in time-dependent models. Viscosity contrasts of six orders of magnitude have been achieved in global mantle flow models [83], however these models were not paleogeographically-driven so could not be as readily compared to the geological record as the models presented here. Modelling magmatic processes in 3D global spherical geometry remains a frontier.

Our results are sensitive to the density of the basal mantle structures, which is a key control on their fractional area ($f_a$). In our preferred Case 2, basal structures were intrinsically 1.43% denser than surrounding mantle. This density is comparable to previous models [e.g., 37, 84], and with that inferred from Earth's free oscillations [up to 1.7%; 85], however, it is larger than those estimated from tidal tomography 0.5% [29]. Further research is required to determine the density of basal mantle structures since seismically-filtered mantle flow models [86] and tomography including Stoneley modes [87] suggest that they could be purely thermal.

We modelled mantle flow forward in time, starting from an uncertain initial condition derived from tectonic reconstructions. A limitation of this approach is that the predicted present-day structure of the lower mantle is similar (up to 78% similarity, Fig 6) rather than identical to the structure of the lower mantle imaged by tomographic models. An alternative approach would consist of reconstructing mantle flow back in time from the present-day structure of the mantle inferred from tomographic models. However, such approaches have so far been limited to the last ~100 Myr [88], and seismically fast structures in tomographic images may constrain plate motions back to 250 Ma at most [67, 89]. We used the only available full-plate tectonic reconstruction extending back to 1 Ga, and considered the original reconstruction [M21; 38] and a version of the reconstruction in which net rotation of the

lithosphere was removed (M21NNR). The latter is appropriate for mantle flow modelling since net rotation of the lithosphere is thought to arise from lateral viscosity variations within the mantle [43, 90]. Indeed, imposed lithospheric net rotation results in induced net rotation of the lower mantle and results in a poor match of lower mantle structures with tomographic models [44]. Nevertheless, lithospheric net rotation is unlikely to be equal to zero; it could be between 0.023˚/Myr [91] and 0.26˚/Myr [92]. Further work is required to develop new tectonic reference frames for paleogeographically-constrained global mantle flow models.

## 5. Conclusion

We assumed that a link exists between upwelling above basal mantle structures and kimberlite magmatism to investigate the relationships between kimberlite eruptions, thick or ancient lithosphere, and lower mantle structure over the last 640 Myr. To investigate the hypothesis that basal mantle structures might have been stationary over time, we considered lower mantle structure to be either stationary and given by tomographic models, or mobile and predicted by reconstructions of past mantle flow over the past billion years. We showed that the present-day structure of the lower mantle predicted by flow models is compatible with that imaged by tomographic models. We created kimberlite potential maps by computing the time-dependent intersection between reconstructed thick or ancient lithosphere and lower mantle structures. We measured the distance of reconstructed kimberlite eruption locations to high kimberlite potential regions, which confirmed that kimberlites are generally close to high kimberlite potential regions, although this result depends on the reference frame being used and the lithosphere extent considered for reconstruction; for the preferred mantle flow model case 50% of kimberlites were within ~250 km of high kimberlite potential areas over 320 Ma and within ~350 km of high kimberlite potential areas over 640 Ma. We found that mobile basal mantle structures better predicted the peak in kimberlite eruption in southern Africa at ~100 Ma than stationary mantle structures. Overall, these results suggest that mobile basal mantle structures are as consistent with kimberlite eruptions as stationary ones. We therefore favour a model in which basal mantle structures have been mobile over time, which is consistent with observations and models suggesting that tectonic plates, subduction zones and mantle plumes have also been mobile over time.

## Supporting information

**S1 Video.**
(MP4)

**S2 Video.**
(MP4)

**S3 Video.**
(MP4)

**S4 Video.**
(MP4)

**S1 Fig.**
(TIF)

## Acknowledgments

This research benefited from discussions with Andrew Macdonald and Khaled Ali from De Beers Exploration. Figs 1B, 3C, 3D, 5, 6, 9E, 9F, 10–12 and S1 Fig were created with Matplotlib [93].

## Author Contributions

**Conceptualization:** Nicolas Flament.

**Data curation:** Nicolas Flament.

**Formal analysis:** Anton Grabreck, Nicolas Flament.

**Funding acquisition:** Nicolas Flament.

**Investigation:** Anton Grabreck, Nicolas Flament.

**Methodology:** Nicolas Flament, Ömer F. Bodur.

**Project administration:** Nicolas Flament.

**Resources:** Nicolas Flament.

**Supervision:** Nicolas Flament, Ömer F. Bodur.

**Validation:** Anton Grabreck, Nicolas Flament.

**Writing – original draft:** Anton Grabreck.

**Writing – review & editing:** Nicolas Flament, Ömer F. Bodur.

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
