## [Decision Letter · Decision Letter 0]

7 Feb 2022

PONE-D-21-39227Global kimberlite prospectivity from reconstructions of mantle flow over the past billion yearsPLOS ONE

Dear Dr. Flament,

Thank you for submitting your manuscript to PLOS ONE. After careful consideration, we feel that it has merit but does not fully meet PLOS ONE’s publication criteria as it currently stands. Therefore, we invite you to submit a revised version of the manuscript that addresses the points raised during the review process.

We look forward to receiving your revised manuscript.

Kind regards,

Shuan-Hong Zhang, PhD

Academic Editor

PLOS ONE

Journal Requirements:

[NF and OFB were supported by Australian Research Council grant LP170100863. This research was undertaken with the assistance of resources from the National Computational Infrastructure (NCI), which is supported by the Australian Government. This research was supported by the Australian Government's National Collaborative Research Infrastructure Strategy (NCRIS), with access to computational resources provided by the National Computational Infrastructure (NCI) through the National Computational Merit Allocation Scheme. Access to NCI was partly supported by resources and services from the University of Wollongong (UOW). Key model results and scripts are available at https://doi.org/10.5281/zenodo.5760115.]

 [NF and OFB were supported by Australian Research Council (https://www.arc.gov.au/) grant LP170100863.]

3. We note that Figures 1, 3, 4, 7, 8, 9, S1 and S2 in your submission contain [map/satellite] images which may be copyrighted. All PLOS content is published under the Creative Commons Attribution License (CC BY 4.0), which means that the manuscript, images, and Supporting Information files will be freely available online, and any third party is permitted to access, download, copy, distribute, and use these materials in any way, even commercially, with proper attribution. For these reasons, we cannot publish previously copyrighted maps or satellite images created using proprietary data, such as Google software (Google Maps, Street View, and Earth). For more information, see our copyright guidelines: http://journals.plos.org/plosone/s/licenses-and-copyright.

a) You may seek permission from the original copyright holder of Figures 1, 3, 4, 7, 8, 9, S1 and S2 to publish the content specifically under the CC BY 4.0 license.  

Additional Editor Comments:

Dear Dr. Nicolas Flament,

Thank you very much for submitting your interesting work to PLOS One. I have received a very detail review on your manuscript. Both reviewer and I found that the topic is interesting and novel and are worthy of publication in PLOS One after a moderate revision. The reviewer has given a numerous comments and suggestions on your manuscript (including those in the attached PDF files), which I believe are very useful for improvement of quality of the manuscript. Please consider all these comments and suggestions carefully during revision.

I am looking forward to receiving your revision manuscript soon.

Yours sincerely,

Shuan-Hong Zhang

Reviewers' comments:

Reviewer's Responses to Questions

**Comments to the Author**

1. Is the manuscript technically sound, and do the data support the conclusions?

Reviewer #1: Yes

2. Has the statistical analysis been performed appropriately and rigorously? 

Reviewer #1: Yes

3. Have the authors made all data underlying the findings in their manuscript fully available?

Reviewer #1: Yes

4. Is the manuscript presented in an intelligible fashion and written in standard English?

Reviewer #1: Yes

5. Review Comments to the Author

Reviewer #1: This work builds up on the hypothesis of Torsvik et al. (2010 Nature) that kimberlites are mostly derived from plumes stemming from the margins of LLSVPs (i.e. seismically defined thermochemical piles above the core-mantle boundary), and assesses the likelihood that the position of mobile thermochemical piles better match the location of Phanerozoic kimberlites compared to fixed thermochemical piles. This is a good idea to test because it is increasingly recognised that LLVSPs are probably not fixed but must be swept around by tectonic processes and mantle convection. To achieve this goal the Authors employ a fluid dynamic model to simulate modification of the thermochemical piles position over the last 640 Myr and identify as zones of kimberlite generation those where the thermochemical piles overlay with the position of cratons (the latter monitored using state-of-the-art plate reconstruction models). They find a good match between kimberlite location and these possible zones of kimberlite generation, and conclude that using mobile rather than fixed thermochemical piles marginally improve the previously proposed hypothesis that most kimberlites are linked to upwellings from these deep mantle structures. They further conclude that this approach could be exploited to identify prospective regions for diamond exploration.

The manuscript is well written and easy to follow. The conclusions are justified by the modelling results. However, I must stress I am not a geodynamic modeller so I could not evaluate the numerical model and some of the boundary conditions. As my expertise is in kimberlite and mantle geochemistry, I have focused my review on the overall approach taken by the Authors while also assessing the soundness of the background information they provide.

I believe the manuscript has one major problem: It relies on craton outlines (those of Merdith et al., 2021) which, to me, have little geological meaning because they extend way beyond known cratonic mantle occurrences and, locally, even into oceanic domains. While the Authors also employ the craton outlines of Artemieva (2006), but hardly discuss this alternative except for one statement, there are more recent and robust definitions of craton (and especially their mantle roots) such as Pearson et al. (2021 Nature), which provide a much better fit for the theme under discussion here.

The problem with employing the craton outlines of Merdith et al becomes clear in figure 9. There is no thick lithosphere (say cratons) in most of Europe (except parts of Ukraine and NW Russia) or in most of US, including all eastern US (e.g., Pearson et al., 2021), which is the opposite of what figure 9 implies (as well as figure 1). A better definition of thick lithosphere will considerably improve the outcome of this work. This issue might seem trivial, but it is not because changing the definition of craton completely modifies the model results. As an example, at L475-476 the Authors state that using the craton outlines of Artemieva (2006) the average distance between predicted and actual position of kimberlites increases significantly, again consistent with the gross overestimation of craton sizes in Merdith et al.

In addition, or perhaps in alternative, if the ultimate aim is to assess diamond prospectivity, it might be better to run the model using as diamond prospective regions only those where the lithosphere is thicker than ~180 km (or ~150 km if the aim is to match the formation of kimberlites) for example based on global tomographic models. This definition of diamond prospective areas is not necessarily confined to cratons but also include thick pericratonic regions (e.g., Limpopo Belt between South Africa and Zimbabwe) where important diamond mines occur (e.g., Venetia). This empirical approach would also circumvent potential problems with craton definition taken from other publications such as that of Merdith and coworkers.

My second major comment will probably sound naïve and reflects my ignorance with fluid dynamic modelling. The Authors start their model at 1 Ga and the conditions are not well specified – i.e. what is the structure of deep-mantle thermochemical piles at this time? Could you please clarify it? They make sure that the model matches the present-day extension and configuration of LLSVPs above the core-mantle boundary to assess model validity, which seems to be logical. However, would not be more insightful to start with the present-day configuration of LLSVPs, which is well established (and not hypothetical) and reconstruct how the position and shape of LLSVP changed backward in time using existing geological constraints (e.g., position of paleo-subduction zones) and mantle flow models? Again, this might sound a naïve comment, but it would be nice to hear the Authors thoughts. It might perhaps be an interesting follow-up work.

Finally, referencing to the literature specific to the origin of kimberlites is often inappropriate and so are some statements – however, I do acknowledge that none of the Authors has a background in kimberlite petrology-geochemistry, so this is not a major issue from my perspective. I have suggested several amendments in the attached pdf and some notable examples follow.

L45. This statement does not make any sense (“the average depth of kimberlite eruptions is 200 km”) and no surprise it comes from obscure Russian literature. Kimberlites originate from depths in excess of 120 km and generally 150 km, this is all we know - e.g., Giuliani and Pearson (2019 Elements) for a concise review (cited later on).

L46. RE superdeep diamonds: also Pearson et al (2014 Nature) and Tschauner et al (2021 Science). However, it should be noted that the evidence supporting a deep kimberlite origin from superdeep diamonds is controversial because these diamonds could have been transported to the base of the lithosphere before being entrained by kimberlite magmas (e.g., Harte and Cayzer, 2007 Phys Chem Min).

L47. The record of kimberlite magmatism is actually episodic not continuous, e.g., Heaman et al., 2019 Elements

L74. I think the only option to link LLSVPs and kimberlites is provided by plumes, either their peripheries or weak ones (see Enrst, 2014 for an assessment of the relationship between LIPs and kimberlites)

L108-110. Unfortunately, the database of kimberlite geochronology by Tappe et al. (2018) has two biases, the first one being geographic areas of more intense exploration for diamonds and the second one being several ages for multiple kimberlite bodies from the same cluster or field (say area). In addition, there might be an inherent preservation bias, which cannot be addressed. I would just remove this statement.

L118-120. Not all kimberlites are located on craton so using Merdith et al. (2021) to obtain craton extensions is not a good idea. Many kimberlites are located in Proterozoic mobile belts and are commonly not diamondiferous. Also, cratons are confined to continental interiors unlike the assessment by Meredith et al. shown in figure 1a.

L364-365. I am not aware of any kimberlite emplaced in Western Australia at this time (~240 Ma). Not the Authors' fault, just one of the several problems with the database of Tappe et al 2018.

L541-544. Additional factors to consider when modelling potential kimberlite formation include lithospheric stress, pre-existing translithospheric structures, plume buoyancy and composition just to name a few.

Finally, it should be kept in mind that not all kimberlites (or related magmas such as those from North China; see Tompkins et al., 1999 Proc 7th Inter Kimb Conf) are necessarily linked to the deep mantle; some might be related to shallower sources including the mantle transition zone (e.g., Kjarsgaard et al., 2017 G-cubed; Chen et al., 2020 Geology) or also the lithospheric mantle (e.g., Giuliani et al., 2015 Nature Comms). This should be noted in the manuscript in my opinion.

Figures 7 and 8 would be easier to follow if the whole continental blocks were shown rather than cratons only. Similarly, the statement at L393-394 is not easy to picture because continental blocks are not shown but only craton outlines.

Additional minor edits and suggested references are included in the attached pdf.

I hope my comments will be helpful and my apologise for the late review (6-Feb-2022)

6. PLOS authors have the option to publish the peer review history of their article (what does this mean?). If published, this will include your full peer review and any attached files.

Reviewer #1: No

---

## [Author Response · Author response to Decision Letter 0]

11 Apr 2022

Our replies to comments are in blue; they include line numbers with reference to the manuscript with tracked changes.

Reviewer #1: This work builds up on the hypothesis of Torsvik et al. (2010 Nature) that kimberlites are mostly derived from plumes stemming from the margins of LLSVPs (i.e. seismically defined thermochemical piles above the core-mantle boundary), and assesses the likelihood that the position of mobile thermochemical piles better match the location of Phanerozoic kimberlites compared to fixed thermochemical piles. This is a good idea to test because it is increasingly recognised that LLVSPs are probably not fixed but must be swept around by tectonic processes and mantle convection. To achieve this goal the Authors employ a fluid dynamic model to simulate modification of the thermochemical piles position over the last 640 Myr and identify as zones of kimberlite generation those where the thermochemical piles overlay with the position of cratons (the latter monitored using state-of-the-art plate reconstruction models). They find a good match between kimberlite location and these possible zones of kimberlite generation, and conclude that using mobile rather than fixed thermochemical piles marginally improve the previously proposed hypothesis that most kimberlites are linked to upwellings from these deep mantle structures. They further conclude that this approach could be exploited to identify prospective regions for diamond exploration.

The manuscript is well written and easy to follow. The conclusions are justified by the modelling results. However, I must stress I am not a geodynamic modeller so I could not evaluate the numerical model and some of the boundary conditions. As my expertise is in kimberlite and mantle geochemistry, I have focused my review on the overall approach taken by the Authors while also assessing the soundness of the background information they provide.

Thank you for this supportive summary of our manuscript.

I believe the manuscript has one major problem: It relies on craton outlines (those of Merdith et al., 2021) which, to me, have little geological meaning because they extend way beyond known cratonic mantle occurrences and, locally, even into oceanic domains. While the Authors also employ the craton outlines of Artemieva (2006), but hardly discuss this alternative except for one statement, there are more recent and robust definitions of craton (and especially their mantle roots) such as Pearson et al. (2021 Nature), which provide a much better fit for the theme under discussion here.

The problem with employing the craton outlines of Merdith et al becomes clear in figure 9. There is no thick lithosphere (say cratons) in most of Europe (except parts of Ukraine and NW Russia) or in most of US, including all eastern US (e.g., Pearson et al., 2021), which is the opposite of what figure 9 implies (as well as figure 1). A better definition of thick lithosphere will considerably improve the outcome of this work. This issue might seem trivial, but it is not because changing the definition of craton completely modifies the model results. As an example, at L475-476 the Authors state that using the craton outlines of Artemieva (2006) the average distance between predicted and actual position of kimberlites increases significantly, again consistent with the gross overestimation of craton sizes in Merdith et al.

In addition, or perhaps in alternative, if the ultimate aim is to assess diamond prospectivity, it might be better to run the model using as diamond prospective regions only those where the lithosphere is thicker than ~180 km (or ~150 km if the aim is to match the formation of kimberlites) for example based on global tomographic models. This definition of diamond prospective areas is not necessarily confined to cratons but also include thick pericratonic regions (e.g., Limpopo Belt between South Africa and Zimbabwe) where important diamond mines occur (e.g., Venetia). This empirical approach would also circumvent potential problems with craton definition taken from other publications such as that of Merdith and coworkers.

We agree that the using the outlines of Merdith et al. (2021) and referring to them as “cratonic” was problematic since they are instead continental blocks inferred to have existed for one billion years. In the revised manuscript, we have implemented the suggestion of the reviewer to consider lithosphere thicker than 150 km, which has been proposed as a first-order proxy for cratonic lithosphere (Pearson et al., 2021). We show the sensitivity of the results to the extent of the continental lithosphere by considering the outlines of Artemieva (2006) for lithosphere older than 2.5 Ga and of Merdith et al. (2021). As a result of this change, we have modified Figs. 1 and 7-12 and the associated text. We also discussed the extent of lithosphere relevant to kimberlite potential mapping in the discussion (L. 938-948). 

My second major comment will probably sound naïve and reflects my ignorance with fluid dynamic modelling. The Authors start their model at 1 Ga and the conditions are not well specified – i.e. what is the structure of deep-mantle thermochemical piles at this time? Could you please clarify it? They make sure that the model matches the present-day extension and configuration of LLSVPs above the core-mantle boundary to assess model validity, which seems to be logical. However, would not be more insightful to start with the present-day configuration of LLSVPs, which is well established (and not hypothetical) and reconstruct how the position and shape of LLSVP changed backward in time using existing geological constraints (e.g., position of paleo-subduction zones) and mantle flow models? Again, this might sound a naïve comment, but it would be nice to hear the Authors thoughts. It might perhaps be an interesting follow-up work.

We have clarified in the discussion that while the initial condition of forward models is uncertain, models starting at the present day and extending back in time are limited to the last ~100 million years (possibly 250 Myr from imaged subducted slabs) and therefore not best suited for the problem at hand (L. 972-979).

Finally, referencing to the literature specific to the origin of kimberlites is often inappropriate and so are some statements – however, I do acknowledge that none of the Authors has a background in kimberlite petrology-geochemistry, so this is not a major issue from my perspective. I have suggested several amendments in the attached pdf and some notable examples follow.

Thank you for these suggestions, which we have considered and addressed as outlined below. Comments in italics were originally made directly on the pdf and have been included here for completeness and transparency.

L14: “or over” � “or through” (L. 18)

L15: “emplacement ages” � “emplacement ages and locations” (L. 20)

L.17: actually the whole Phanerozoic if one follows the original hypothesis of Torsvik and coworkers � we disagree. Torsvik et al. (2010) established a relationship between kimberlites and LLSVPs for the last 320 Myr, then assumed it for the rest of the Phanerozoic.

L. 40: Pearson et al., 2021 Nature is probably a better reference for this statement � added reference to Pearson et al. (2021) (L. 58)

L42: Some of these references are either a bit obscure or not well suited for this statement. Better Stachel and Harris (2008 Ore Geol Rev), Helmstaedt and Gurney (1995 J Geochem Explor), Kjarsgaard et al. (2019, Elements) � references updated; thank you for these suggestions (L. 60).

L.43: ascending � change made (L. 61).

L.43: Inappropriate references. Better Sparks et al. (2006 JVGR) and Canil and Fedortchouk (1999 EPSL) who actually calculated ascent speeds � references updated; thank you for these suggestions (L. 60).

L45. This statement does not make any sense (“the average depth of kimberlite eruptions is 200 km”) and no surprise it comes from obscure Russian literature. Kimberlites originate from depths in excess of 120 km and generally 150 km, this is all we know - e.g., Giuliani and Pearson (2019 Elements) for a concise review (cited later on).

We have changed this statement to: “Kimberlites originate from depths in excess of 120-150 km [12], forming from melts that may pool from up to ~300 km depth [13]” (L. 63-64).

L46. RE superdeep diamonds: also Pearson et al (2014 Nature) and Tschauner et al (2021 Science). 

We added Pearson et al (2014 Nature) and Tschauner et al (2018 Science) (L. 65).

However, it should be noted that the evidence supporting a deep kimberlite origin from superdeep diamonds is controversial because these diamonds could have been transported to the base of the lithosphere before being entrained by kimberlite magmas (e.g., Harte and Cayzer, 2007 Phys Chem Min).

We stated “although these diamonds could have been transported to the base of the lithosphere before being entrained by kimberlite magmas” and cited Harte and Cayzer (2007 Phys Chem Min) (L. 65-67).

L47. The record of kimberlite magmatism is actually episodic not continuous, e.g., Heaman et al., 2019 Elements

“continuous” has been changed to “episodic” and references have been updated (L. 68).

L. 48: incomplete reference in the reference list � reference updated (L. 68).

L.52: Another rather obscure reference. Perhaps better Giuliani and Pearson 2019 which include a map of kimberlite distribution � references updated (L. 72-73).

L.55: (see also L. 17) the relationship was established back to 320 Ma, not back to 540 Ma

L. 61: If not mistaken, the best assessment of the size of LLSVPs is that of Cottaar and Lekic (2016 Geophys J Intern) � we disagree. The cluster analysis method used by Cotaar and Lekic leads to much larger LLSVP volumes than derived from individual tomographic models. 

L.64: comma added "e.g. ,” (L. 98 and elsewhere throughout the manuscript).

L. 70: also the various papers of Campbell and Griffith, e.g., 1990 EPSL, which come much earlier than Courtillot et al. � we added a reference to Campbell & Griffiths 1990 (L. 104).

L.72: also Ernst (2014): Large Igneous Provinces � we added a reference to Heaman & Kjarsgaard (2000) (L. 106).

L.73: [on average from ~ 200 km depth; 3] � replaced with [from 120-300 km depth; 12, 13] (L. 107).

L74. I think the only option to link LLSVPs and kimberlites is provided by plumes, either their peripheries or weak ones (see Enrst, 2014 for an assessment of the relationship between LIPs and kimberlites) 

We agree that heat must be advected from the deep mantle to the surface by mantle upwelling, however, we note that kimberlites are not systematically associated with large igneous provinces.

L.78: In this context the work of Conrad et al. (2013 Nature) is probably relevant, i.e. LLSVP might be fixed only over no more 100s Myr � Conrad et al. (2013) did not present geodynamic models. They argued that LLSVPs are stationary and control mantle flow based on an analysis of surface plate motions over time and present-day mantle flow. This work is at odds with the statement: “global geodynamic models have revealed that similar hot basal mantle structures can form in response to the history of surface plate motions and plate subduction”

L.80: see also Giuliani et al (2021 PNAS) for an evaluation of this model in light of existing geochemical constraints for kimberlites worldwide. Not all kimberlites are related to deep-mantle structure and when they are not the isotopic features seem to be distinct from those of kimberlites that are linked to deep-mantle upwellings � we agree, and we have made that point in the discussion (L. 924-933).

L.91: and kimberlite emplacement ages � change made (L. 145).

L108-110. Unfortunately, the database of kimberlite geochronology by Tappe et al. (2018) has two biases, the first one being geographic areas of more intense exploration for diamonds and the second one being several ages for multiple kimberlite bodies from the same cluster or field (say area). In addition, there might be an inherent preservation bias, which cannot be addressed. I would just remove this statement.

We have deleted the statement.

L. 117: Please also check the recent Pearson et al. (2021 Nature), which include geophysical and petrological constraints � this point has guided our revision and has resulted in significant changes to the text and to the figures

L118-120. Not all kimberlites are located on craton so using Merdith et al. (2021) to obtain craton extensions is not a good idea. Many kimberlites are located in Proterozoic mobile belts and are commonly not diamondiferous. Also, cratons are confined to continental interiors unlike the assessment by Meredith et al. shown in figure 1a.

We agree that using Merdith et al. (2021) to obtain craton extensions is not a good idea. We now consider lithosphere thicker than 150 km for the main analysis, and for a sensitivity study we also consider regions with a tectonothermal age greater than 2.5 Ga (Artemieva, 2006) and tectonic blocks inferred to have existed for at least one billion years (Merdith et al., 2021).

L.120-122: “this statement is out of context here, please move it to the previous paragraph” � change made (L. 165-166).

L. 231: Or is this just a comparison of tomographic models? Please clarify for the uneducated reader (such as myself) � change made (L. 372-373).

L. 334: please remind the reader the parameters employed in Case 2 - I had to go back to Table 1 to refresh my memory at this point. � change made (L. 498-499).

L. 349-350: please specify the age of these kimberlites and/or the panels of figure 7 when these kimberlites are specifically shown � change made (L. 516-519).

L364-365. I am not aware of any kimberlite emplaced in Western Australia at this time (~240 Ma). Not the Authors' fault, just one of the several problems with the database of Tappe et al 2018.

The Roper kimberlite field was emplaced in the Northern Territory (rather than in Western Australia). According to Hutchison (2012): “Age ranges of NT Roper field kimberlites from 65-250 Ma”. Tappe et al. (2018) list the Packsaddle-01 kimberlite as 238 Myr old (method: Rb-Sr on Phlogopite/biotite; no reference given).

L. 475-476: consistent with the likely overestimation of craton size in Meredith et al � we have modified the text to reflect that this has been changed

L. 489: perhaps move Figure 13 to appendix � we agree and have moved Figure 13 to the supplement (S1 Figure)

L.497-499: this is an excellent idea and something already under consideration in the diamond industry; however, a tighter definition of diamond-bearing lithospheric mantle is needed compared to the 'cratons' employed in the manuscript.

We now consider lithosphere thicker than 150 km.

 it should also be considered the triggers of kimberlite magmatism because plumes are not continuously produced along LLSVPs

We agree that the mechanism linking kimberlite magmatism to LLSVPs should be investigated in the future. In this study, the link is assumed as in Torsvik et al. (2010).

L.516: also Conrad et al. (2013 Nature) � the analysis by Conrad et al. (2013) is debated (see Rudolph and Zhong, 2013, Nature); we added a reference to Dziewonski et al. (2010) (L. 906).

L.533: see also Kjarsgaard et al. (2017 G-cubed) or Chen et al. (2020 Geology) � we cited Chen et al. (2020 Geology) (L. 919) who explained why the model proposed by Kjarsgaard et al. (2017 G-cubed) does not work.

L. 536: Also Tappe et al. (2013 EPSL) and Tovey et al. (2021 Lithos) who both link recent kimberlite magmatism in the Slave craton to past subduction events � we cited Tappe et al. (2013 EPSL) (L. 924)

L541-544. Additional factors to consider when modelling potential kimberlite formation include lithospheric stress, pre-existing translithospheric structures, plume buoyancy and composition just to name a few.

We mentioned that lithospheric controls on potential kimberlite formation were not considered (L. 917).

Finally, it should be kept in mind that not all kimberlites (or related magmas such as those from North China; see Tompkins et al., 1999 Proc 7th Inter Kimb Conf) are necessarily linked to the deep mantle; some might be related to shallower sources including the mantle transition zone (e.g., Kjarsgaard et al., 2017 G-cubed; Chen et al., 2020 Geology) or also the lithospheric mantle (e.g., Giuliani et al., 2015 Nature Comms). This should be noted in the manuscript in my opinion.

We have modified this paragraph, including citations to Chen et al. (2020 Geology) (L. 919) and to Giuliani et al. (2015 Nature Communications) (L. 922).

Figures 7 and 8 would be easier to follow if the whole continental blocks were shown rather than cratons only. Similarly, the statement at L393-394 is not easy to picture because continental blocks are not shown but only craton outlines.

We now show present-day coastlines as well as lithosphere thicker than 150 km on Figures 7 and 8.

Additional minor edits and suggested references are included in the attached pdf.

These have been addressed above (in italics).

I hope my comments will be helpful and my apologise for the late review (6-Feb-2022)

Thank you for these comments that have taken the manuscript to a new level in terms of referencing and have given us the opportunity to improve the definition of the extent of continental lithosphere relevant to kimberlite exploration.

---

## [Decision Letter · Decision Letter 1]

22 Apr 2022

Mapping global kimberlite potential from reconstructions of mantle flow over the past billion years

PONE-D-21-39227R1

Dear Dr. Flament,

We’re pleased to inform you that your manuscript has been judged scientifically suitable for publication and will be formally accepted for publication once it meets all outstanding technical requirements.

Kind regards,

Shuan-Hong Zhang, PhD

Academic Editor

PLOS ONE

Additional Editor Comments (optional):

Dear Dr. Flament,

Thank you very much for revising the manuscript and addressed all the comments and suggestions clearly. After evaluating from the previous reviewers and my reading through of your manuscript, I am pleased to confirm that your paper has been accepted for publication in PLOS ONE.

Thank you for choosing PLOS ONE to plublish your interesting work.

Best wishes,

Shuan-Hong Zhang

Reviewers' comments:

Reviewer's Responses to Questions

**Comments to the Author**

1. If the authors have adequately addressed your comments raised in a previous round of review and you feel that this manuscript is now acceptable for publication, you may indicate that here to bypass the “Comments to the Author” section, enter your conflict of interest statement in the “Confidential to Editor” section, and submit your "Accept" recommendation.

Reviewer #1: All comments have been addressed

2. Is the manuscript technically sound, and do the data support the conclusions?

Reviewer #1: (No Response)

3. Has the statistical analysis been performed appropriately and rigorously? 

Reviewer #1: (No Response)

4. Have the authors made all data underlying the findings in their manuscript fully available?

Reviewer #1: (No Response)

5. Is the manuscript presented in an intelligible fashion and written in standard English?

Reviewer #1: (No Response)

6. Review Comments to the Author

Reviewer #1: (No Response)

7. PLOS authors have the option to publish the peer review history of their article (what does this mean?). If published, this will include your full peer review and any attached files.

Reviewer #1: **Yes: **Andrea Giuliani

---

## [Editor Report · Acceptance letter]

25 May 2022

PONE-D-21-39227R1 

Mapping global kimberlite potential from reconstructions of mantle flow over the past billion years 

Dear Dr. Flament:

I'm pleased to inform you that your manuscript has been deemed suitable for publication in PLOS ONE. Congratulations! Your manuscript is now with our production department. 

Kind regards, 

on behalf of

Dr. Shuan-Hong Zhang 

Academic Editor

PLOS ONE